# Prognostic Biomarkers of Cell Proliferation in Colorectal Cancer (CRC): From Immunohistochemistry to Molecular Biology Techniques

**DOI:** 10.3390/cancers15184570

**Published:** 2023-09-15

**Authors:** Aldona Kasprzak

**Affiliations:** Department of Histology and Embryology, University of Medical Sciences, Swiecicki Street 6, 60-781 Poznan, Poland; akasprza@ump.edu.pl; Tel.: +48-61-8546441; Fax: +48-61-8546440

**Keywords:** colorectal cancer, cell proliferation, genetic and epigenetic markers, cell cycle-related antigens, prognostic markers, cyclins, PCNA, Ki-67 antigen, ncRNAs, immunohistochemistry, qRT-PCR, RNA/DNA sequencing

## Abstract

**Simple Summary:**

This review aims to shed light on the proliferative markers important in the everyday clinical management of colorectal cancer (CRC), ranging from simple methods of assessing cellular proliferation (e.g., DNA ploidy, BrdUrd/IdUrd/tritiated thymidine binding index) to the use of immunohistochemistry (IHC) and modern molecular biology techniques (e.g., qRT-PCR, in situ hybridization, RNA/DNA sequencing) for the detection of genetic and epigenetic markers. Among the examined markers, the prognostic utility was demonstrated for aneuploidy and the overexpression of IHC markers (e.g., TS, cyclin B1, and D1, PCNA, and Ki-67). Classical genetic markers of prognostic significance mostly comprise mutations in commonly examined genes such as *APC*, *KRAS/BRAF*, *TGF-β*, and *TP53*. Chromosomal markers include CIN and MSI, while CIMP is indicated as a potential epigenetic marker with many other candidates such as *SERP*, *p14*, *p16*, *LINE-1*, and *RASSF1A.* Modern technology-based approaches to study non-coding fragments of the human genome have also yielded some candidates for CRC prognostic markers among the lncRNAs (e.g., SNHG1, SNHG6, MALAT-1, CRNDE) and miRNAs (e.g., miR-20a, miR-21, miR-143, miR-145, miR-181a/b). With growing knowledge of the human genome structure and the rapid development of molecular biology techniques, it is hoped that a panel of reliable prognostic markers could improve the assessment of survival as well as allow for the better estimation of the treatment outcomes for CRC patients.

**Abstract:**

Colorectal cancer (CRC) is one of the most common and severe malignancies worldwide. Recent advances in diagnostic methods allow for more accurate identification and detection of several molecular biomarkers associated with this cancer. Nonetheless, non-invasive and effective prognostic and predictive testing in CRC patients remains challenging. Classical prognostic genetic markers comprise mutations in several genes (e.g., *APC*, *KRAS/BRAF*, *TGF-β,* and *TP53*). Furthermore, CIN and MSI serve as chromosomal markers, while epigenetic markers include CIMP and many other candidates such as *SERP*, *p14*, *p16*, *LINE-1*, and *RASSF1A*. The number of proliferation-related long non-coding RNAs (e.g., SNHG1, SNHG6, MALAT-1, CRNDE) and microRNAs (e.g., miR-20a, miR-21, miR-143, miR-145, miR-181a/b) that could serve as potential CRC markers has also steadily increased in recent years. Among the immunohistochemical (IHC) proliferative markers, the prognostic value regarding the patients’ overall survival (OS) or disease-free survival (DFS) has been confirmed for thymidylate synthase (TS), cyclin B1, cyclin D1, proliferating cell nuclear antigen (PCNA), and Ki-67. In most cases, the overexpression of these markers in tissues was related to worse OS and DFS. However, slowly proliferating cells should also be considered in CRC therapy (especially radiotherapy) as they could represent a reservoir from which cells are recruited to replenish the rapidly proliferating population in response to cell-damaging factors. Considering the above, the aim of this article is to review the most common proliferative markers assessed using various methods including IHC and selected molecular biology techniques (e.g., qRT-PCR, in situ hybridization, RNA/DNA sequencing, next-generation sequencing) as prognostic and predictive markers in CRC.

## 1. Introduction

Colorectal cancer (CRC) remains a major medical challenge worldwide, ranking third in prevalence and second among cancer-related death causes [1,2,3,4]. The high mortality rate persists in European countries, but also affects several other regions around the world such as the Caribbean, East Asia (China), and South America (Uruguay), indicating a continuously high incidence as well as lackluster detection and treatment methods [3]. An increase can also be observed in CRC incidence in younger people (under 50 years of age), with those predisposed to CRC generally classified as ‘medium’ and ‘high’ risk groups [1,3,5,6]. Moreover, the incidence of CRC is positively correlated with the levels of the human development index (HDI) [6]. 

The development of CRC is a multistage process. The numerous genetic alterations in CRC are reflected in morphological features that can be visualized by various molecular techniques [7,8,9]. A significant role in tumor initiation, growth, and metastasis is now attributed to cancer/tumor-initiating cells (CICs/TICs) or cancer stem cells (CSCs), which are capable of self-renewal and differentiation. Numerous studies support the ‘CSC hypothesis’, in which the essence of carcinogenesis is progressive colonic SC overpopulation. Research is ongoing into the biology of these cells, the identification of their molecular markers, and the mechanisms of CSC proliferation, differentiation, and resistance to treatment in CRC [10,11,12,13,14,15].

There are several theories regarding the sequence of events in the formation of CRC [16]. The first pathway of the ‘adenoma–carcinoma sequence’ describes a sequence of morphological alterations, from hyperplasia through dysplasia to the formation of malignant, invasive foci [7,17]. Pre-cancerous lesions, in this case, comprise adenomatous polyps [7], with the ‘adenoma–carcinoma sequence’ concept supplemented by early dysplastic lesions, known as aberrant crypt foci (ACF) [18,19]. The second theory of CRC formation, known as the mutator pathway, took its origin from the 1992 discovery of genetic alterations in patients with Lynch syndrome (LS), also known as hereditary non-polyposis CRC (HNPCC) [20].

Approximately 15% of CRC arises from genetic alterations. Several syndromes can be distinguished in the etiology of CRC, associated with a high lifetime risk of CRC due to the inheritance of mutations in a single gene. Specific ‘Mendelian’ CRC syndromes include familial adenomatous polyposis (FAP), with gene mutation of the adenomatous polyposis coli (*APC*) gene, LS genes (*MSH2*, *MLH1, MSH6*, *PMS2*), Peutz–Jeghers syndrome (*LKB1/STK11*), juvenile polyposis (*SMAD4, BMPR1A*), *MUTYH*-associated polyposis, and hereditary mixed polyposis (*GREM1*). All of these conditions, except for *MUTYH*-associated polyposis, are inherited in a dominant manner. However, there is a recessive version of HNPCC in which both copies of one of the DNA mismatch repair (*MMRs)* genes are mutated (reviewed in [21]).

A third theory of CRC development, the serrated pathway or hyperplastic polyp-carcinoma sequence, considers hyperplastic polyps (HPs) together with a subgroup of serrated polyps (SPs) as precursors of CRC [22]. It is now recognized that up to 10–30% of CRC cases arise through this alternative pathway, characterized by its genetic and epigenetic profile [23,24,25,26,27].

From a clinical perspective, people with LS or colorectal polyposis syndromes are at the most significant risk of developing CRC. LS accounts for 1–3% of all cases, with people affected with this syndrome characterized by an absolute CRC risk ranging from 30 to 70% [28]. Increased risk also applies to people with colorectal adenomatous polyps, inflammatory bowel disease, a history of CRC, or cases of this cancer in close family members under the age of 50. Low to moderate risk of developing CRC applies to virtually the entire population, associated with age over 50 years and the consequences of an unhealthy lifestyle leading to obesity and other metabolic disorders, resulting in the production of a number of tumor-promoting proteins [12,29]. Obesity prevention, especially among the young human population, is an important preventive factor for CRC. A recent meta-analysis showed that overweight and obesity may be more potent risk factors for CRC and, possibly, other cancers than the previous epidemiological studies suggested [30].

Together with changes in chromatin structure and DNA methylation, gene mutations in CRC lead to the dysregulation of signaling pathways responsible for cell proliferation, apoptosis, metabolism, differentiation, and survival [16,21,31,32].

The ’adenoma–carcinoma sequence’ is mainly characterized by a loss of proliferation control. In turn, in the serrated neoplasia pathway, a failure of apoptosis mechanisms is the most characteristic factor. However, asymmetric proliferation (shift of the zone of proliferation from the base to the lateral side) is typical of the architecturally distorted serrated crypt, a characteristic of sessile serrated lesions (SSLs) [22].

CRC is a typically malignant tumor characterized by genetic/epigenetic mutations in mutator genes (i.e., genes whose alterations accelerate mutations in other genes). However, the molecular and cellular alterations associated with the immortality (abnormal maintenance of proliferation) and autonomy of colorectal cells, as with other malignancies, remain unknown (reviewed in [33]). Studies also suggest that the mechanism linking abnormalities at the genetic (e.g., *APC* mutations) and cellular level (e.g., hyperplasia, dysplasia) between tumor initiation to metastasis is the excessive number of colonic CSCs. It also considers the symmetrical division of CSCs as an essential mechanism driving tumor growth, which may have therapeutic implications for patients with advanced CRC [34].

Due to the above, searching for optimal methods to evaluate tumor proliferation and for more sensitive markers with potential prognostic significance seems crucial. A prognostic factor is a variable that indicates the predicted natural course of the disease and can be used to estimate the chance of recovery or the likelihood of recurrence. Prognostic significance is particularly relevant to progression-free survival (PFS) and overall survival (OS). Prognostic factors are classified into tumor-related, host-related, and environmental [35,36].

The aim of this article was to review the most common proliferative markers assessed by various methods including immunohistochemistry (IHC) and selected molecular biology techniques (e.g., qRT-PCR, in situ hybridization, RNA/DNA sequencing, next-generation sequencing) as prognostic and predictive CRC markers.

## 2. Molecular Mechanisms of Colorectal Cancerogenesis

At the core of the classical pathway of CRC development (‘adenoma–carcinoma sequence’) are genetic alterations of several suppressor genes such as *APC,* responsible for the development of FAP, and a gene known as colorectal mutant cancer protein (*MCC)* [37,38]. *APC* mutations usually result in activation of the canonical Wnt pathway [39]. Further genes include deleted in colorectal cancer *(DCC)*, encoding members of the CAM immunoglobulin family of adhesion proteins [40], similar to neural cell adhesion molecules (NCAMs) [41]. Its product acts as a netrin 1 receptor [42] and is often silenced in CRC through the loss of heterozygosity or epigenetic mechanisms [43]. *TP53* and the *K-ras (K-RAS, KRAS)* and *BRAF* protooncogenes are also implicated in the development of CRC, playing a role in the MAPK signaling pathway [5,9,44]. However, further studies have indicated that alterations in the three ‘classical’ carcinogenesis genes (*APC*, *K-RAS*, *TP53*) affect only about 10% of CRC, as this cancer is characterized by considerable genetic heterogeneity [23].

Thus, according to the conventional pathway theory of colorectal carcinogenesis, the first step involves *APC* changes, resulting in increased cell proliferation and polyp formation. In the next step, genetic alterations of *K-RAS* result in further clonal tissue proliferation and increased polyp size. This is followed by polyp proliferation due to *DCC* mutations. *TP53* mutations with telomerase activation are reported in approximately 70% of CRC cases. Mutation in the *TP53* leads to malignancy, resulting in metastasis to the surrounding tissues and distant organs [5,16,44].

Nowadays, it is known that at the molecular level, chromosomal instability (CIN) (~70–85%), extensive DNA methylation known as CpG island methylator phenotype (CIMP) (~17% CRC), and microsatellite instability (MSI) (~15% CRC) are the most common factors in the classical pathway of colorectal carcinogenesis [9,17,23,32]. The presence of CIN in tumors results in the accumulation of mutations in oncogenes and tumor suppressor genes (*APC*, *TP53, KRAS*, and *BRAF*). However, more than 24 mutated genes have currently been associated with CRC [45], with the number possibly higher due to the development of modern testing techniques including single-cell next-generation sequencing (NGS) [46].

In contrast to CIN, in MSI, morphological changes are associated with minor aneuploidy, with LS serving as a typical example (3% CRC) [20]. The main characteristic of these lesions is the mutation of *MMR* genes, namely *MLH1*, *MSH2*, *MSH6*, and *PMS2*, with no congenital polyps in the CRC development sequence. The process of neoplastic transformation, however, is similar to that in the CIN pathway (i.e., with a prior development of adenoma (AD)). On the LS (mutator pathway), the development of CRC can occur through (1) sporadic adenomas that acquire secondary *MMR* deficiency (dMMR); (2) flat intramucosal lesions that arise directly from dMMR crypts; (3) LS-specific adenomas that arise from flat lesions as a result of secondary *APC* mutations [47]. A subgroup of hypermutating carcinomas that do not show MSI features has also been demonstrated. In addition, families with oligopolyposis and MS stable (MSS) at a young age but without *APC* or *MYTYH* (*MYH*) mutations have been identified. Investigations of further mechanisms responsible for the hypermutation revealed germline exonuclease domain (*EDM*) mutations of *POLE* and *POLD1* genes, associated with a high risk of multiple ADs and CRC, resulting in a condition known as polymerase proofreading-associated polyposis (PPAP). Somatic *POLE EDMs* have also been found in sporadic CRC, although very few *POLD1* somatic *EDMs* have been described [48].

In the third concept, the so-called serrated pathway (‘hyperplastic polyp–carcinoma sequence’), HPs, together with a subgroup of serrated polyps (SPs), have been recognized as precursors of CRC [22]. It is now known that as many as 10-30% of CRCs arise through this alternative pathway, characterized by their own genetic and epigenetic profile [23,24,25,26,27]. The most recent classification of serrated colorectal lesions (formerly known as sessile serrated polyp/adenoma) describes them as precursors of various molecular CRC subtypes [22,49]. In these lesions, hypermethylation of cytosine residues within CpG islands can sometimes be observed. Point mutations of *B-Raf* protooncogene serine/threonine kinase (*BRAF)*, promoter methylation of multiple genes, and MSI have also been described in the serrated pathway [24]. In turn, the molecular mechanisms of the CIMP pathway are not well understood. These cancers are characterized by a poorer prognosis but can be detected earlier, as aberrant DNA methylation is already present [50]. Morphologically, the CIMP pathway is associated with lesions with a characteristic microscopic ‘serrated’ mucosal edge structure, previously identified as hypertrophic benign polyps. These polyps are currently known as sessile serrated adenomas/polyps (SSA/Ps) and have been recognized as major precursor lesions for CRC. They can arise from HPs or de novo from normal mucosa [24,25]. Serrated polyposis syndrome (SPS) is also a risk factor for CRC, characterized by large and multiple serrated polyps throughout the colon. The most common genetic variants associated with CRC susceptibility in SPS patients are rs4779584-GREM1, rs16892766-EIF3H, and rs3217810-CCND2 [51].

Interestingly, the MSS/CIMP-negative subset has been shown to evolve through the classical ‘adenoma–carcinoma sequence’. In contrast, the MSI/CIMP-positive and MSS/CIMP-positive subsets often develop through the ‘serrated pathway’ [23]. As indicated by a recent cohort study (~30,000 participants) evaluating 40 established CRC susceptibility subtypes, common genetic variants play a potential role in conventional and serrated CRC pathways. The occurrence of different sets of variants for these two pathways demonstrates the etiological heterogeneity of CRC [52]. It should be noted that a third concept of CRC development (in addition to the conventional tubular/villous adenoma–carcinoma and the serrated adenoma–carcinoma pathways) has been proposed (although much less common), namely, cancer formation in the mucosal domain of gut-associated lymphoid tissue (GALT) [53].

There are currently four molecular subtypes of CRC, the so-called consensus molecular subtypes (CMS) (i.e., CMS1—immunological, CMS2—canonical, CMS3—metabolic, and CMS4—mesenchymal). Considering the clinical features, biology, and gene signatures of colon cancer subtypes (CCS), the CCS3 subtypes, whose precursors are SSAs, have the worst prognosis (defined by the shortest disease-free survival, DFS). Using the CRCassigner signature to classify the TCGA dataset, they are known as the stem-like and transit-amplifying (TA) (including cetuximab-resistant TA) subtypes [54].

Among the numerous molecular markers of CRC (more than 100 differentially expressed), most are overexpressed during tumorigenesis. Functionally, they are involved in various biological signaling pathways including those related to cell proliferation [55].

## 3. Cellular Proliferation Models versus Colorectal Carcinogenesis Theories

Complex cell cycle (mitotic cycle) mechanisms regulate proliferation, survival, and death. The processes and factors involved in cell cycle regulation in mammalian cells in physiology and tumorigenesis have been well-characterized in numerous reviews [56,57,58,59]. The cell cycle is primarily driven by the activation of serine/threonine cyclin-dependent kinases (Cdks) by cyclins and the phosphorylation and dephosphorylation of Cdks [56,57,60]. In human cells, there are 20 different Cdks and about 30 cyclin genes [57,61], which, in addition to participating in the cell cycle process, are also involved in transcription and pre-mRNA splicing [62]. In addition to Cdks, which drive cell passage through the phases of the cell cycle, there are also kinase inhibitors that regulate it and prevent it from progressing. The concentration of Cdks in the cell is constant, while the concentration of cyclins varies according to the cycle phase. The most significant role in cell cycle progression and its timely and precise regulation is attributed to the ubiquitin–proteasome system [59].

Cell cycle genes encoding proteins that stimulate the cell cycle are known as protooncogenes, and those that inhibit the cell cycle are the suppressor genes. In a cancer cell, genetic changes result in the conversion of protooncogenes to oncogenes, and the loss of function of some suppressor genes. This leads to a steady production of proteins that induce cell division (products of oncogenes) and a deficiency of proteins that inhibit this process (suppressor genes products). According to the clonal theory of oncogenesis, tumor formation starts from a single cell. Furthermore, there is a close relationship between tumor development and the inhibition of apoptosis, which ensures cell immortality [63]. Dysregulation proliferation, apoptosis, and autophagy factors also include altered Ca^2+^ transmission [64].

According to the somatic mutation theory (SMT) of carcinogenesis, external cancer-causing agents (e.g., environmental, chemical, radiation, carcinogens) damage the DNA of a single cell, leading to the generation of mutations. These, in turn, through successive rounds of cell proliferation and clonal selection, drive the process of carcinogenesis. In contrast, tissue organization field theory (TOFT) recognizes that proliferation and motility are the default states of all cells [65]. Among the current 10 hallmarks of cancer, in addition to ‘genomic instability and mutations’, ‘non-mutational epigenetic reprogramming’, and ‘polymorphic microbiomes’ are ‘sustaining proliferative signaling’, ‘enabling replicative immortality’, and ‘resisting cell death’ [66].

Unlike normally differentiating cells, cancer cells can enter the proliferation or tumorigenesis pathway from the G0 to G1 phase (the G0 repose model) [67]. This hypothesis assumes that in a tumor, there are non-proliferating cells in the G0 phase, forming a resting compartment (quiescent, Q). The fate of these cells was dual, either re-entering the cycle through the G1 phase with growth factors, cytokines, oxygenation, and nutrients or cell death (after exiting the G0 phase and Q compartment) [67,68]. In tumor tissues, the proliferative process predominates, resulting in a greater withdrawal (especially in the absence of nutrients, hypoxia) of cells from the G1 to G0 phase. This greater number of cells in the G0 phase is characteristic of solid tumors including CRC. Another proliferation model (the growth retardation model) assumed that, under unfavorable conditions, the withdrawal of cells to the G0 phase could occur in any stage of the cycle, not only in the G1 phase [69]. A few years later, a multilevel model of cancer cell proliferation, known as the proliferation plane model, was proposed [70]. It assumes the existence of different subpopulations of cells that differ in growth rate, rates of cycling, and recruitment to the cycle within a single population. A modification of this model is the so-called Wilson’s integrated tumor growth model, which also assumes different subpopulations of cells in the tumor but also various factors affecting tumor growth (e.g., differentiation, apoptosis, tumor microenvironment (TME) factors) [68]. The advantage of both models is the potential prognostic and predictive significance of a subpopulation of slowly proliferating cells in the tumor and the depiction of the molecular mechanisms controlling the division cycle of tumor cells. These cells may include CSCs that reside in the G0 phase (like SCs of normal tissues) or proliferate very slowly.

The assessment of proliferation in CRC, especially in a prognostic and diagnostic context, has been the focus of scientists and clinicians for a number of years. The difficulty in interpreting many findings in this area is related to the enormous heterogeneity of the tumor in terms of genotype, phenotype, morphology, and cell metabolism [46]. Interestingly, while epithelial cells in the large intestine have a longer lifespan and proliferate slower than in the small intestine (5–21 days versus 3–4 days) [71,72], colon cancer is far more common than small intestinal cancers (10% versus <1% of all cancers) [4]. This discrepancy between proliferation characteristics and the risk of uncontrolled, malignant tissue transformation is called ‘the proliferation paradox’ [73]. For example, patients with FAP are ∼30 times more likely to develop CRC than duodenal cancer. Studies by Tomasetti and Vogelstein suggest that this occurs as there are ∼150 times more SC divisions in the colon than in the duodenum. The risk of CRC would be very low (even with the *APC* mutation) if the SCs of the colonic epithelium were not constantly dividing [74]. Thus, both SCs and non-SCs, which may differentiate into an SC-like cell phenotype, are suggested to be involved in colon carcinogenesis [75]. In the ‘top–down’ model of CRC heterogeneity involving intestinal SCs (ISCs), tumor initiation would start at the top of the crypt, where *APC*-mutated cells are observed and spread laterally and downward toward the normal crypt [76]. The second model of carcinogenesis is the spread of cancer ‘from the bottom up.’ In patients with a familial predisposition to *APC* mutations, dysplastic lesions have been observed on the tissue surface and then within individual crypts. Hence, this direction of lesion spread involving ISCs is not excluded [77].

The tissue heterogeneity of CRC is explained in two ways, namely (1) in the CSC model and (2) in the clonal evolution model. In the former, tumor cells are organized hierarchically. Some of them are CSCs, which retain the ability to proliferate, while their progeny ‘differentiate’ into non-proliferating lineages [10,13]. It was in colon cancer that the different subpopulations of CSCs/CICs/TICs, which are responsible for the different stages of CRC development in primary CRC (pCRC), were first distinguished [14]. Previous studies indicated that in the progression from normal to the mutated epithelium of AD, aldehyde dehydrogenase 1 (ALDH1)-positive cells restricted to the normal crypt bottom increased in number and became distributed further up the crypt. This marker was therefore found to be a favorable marker of CSCs responsible for tumor progression [12]. The role of CSCs with a leucine-rich repeat-containing G-protein-coupled receptor 5 (Lgr5)(+) phenotype, essential for tumor growth and metastasis formation (e.g., in the liver), has also been demonstrated in growing CRC tumor tissues [78,79,80]. Genetic experiments have confirmed that these dynamic CSCs are at the top of the hierarchy of human CRC cells, and this organization resembles that of the normal colonic epithelium [78].

Interestingly, ablation of Lgr5(+) CSCs did not inhibit the growth of the primary tumor, as Lgr5(+) CSCs were continuously replenished by proliferative Lgr5(−) cancer cells, but resulted in reduced liver metastasis (CRLM) [79]. There has long been research evaluating other colon markers of CSCs and the mechanisms controlling the rate of division and self-renewal, which may confer tumor growth and be the cause of chemoresistance [81,82]. In a rat model, it has been shown that only 1 in 25 cells, or 1 in 262 cells, have the characteristics of CSCs in the whole CRC cell population [11,12]. Moreover, the CSC480 CRC stem cell line exhibited an elevated expression of CSC markers such as CD44, ALDH1, and Sox2 compared to the grade 3–4 colon adenocarcinoma cell line (SW480). In addition, the quiescent cells were detected in a heterogeneous tumor cell population using the proliferation marker 5-ethynyl-2’-deoxyuridine (EdU) and a label-retaining cell protocol. Most of the normal fetal human colon epithelial cell line (FHC) resided in this quiescent state. These cells are characterized by extremely slow cell division, as evidenced by the increased expression of ALDH1 compared to other cell lines. In addition, elevated ATP-binding cassette superfamily G member 2 (ABCG2) expression was also present in the FHC cells compared to the SW480 and CSC480 cells. This may support reports that quiescent cells are resistant to chemotherapy (CTx) [82].

The clonal evolution model assumes that genetic and epigenetic changes occur over time in individual cancer cells and that if such changes confer a selective advantage, they will allow particular cancer cell clones to compete with other clones. Clonal evolution can lead to genetic heterogeneity, resulting in phenotypic and functional differences between cancer cells within a single patient [13]. While initial studies suggested that colorectal tumors were monoclonal, later research has shown that the majority (up to 76%) of human early microadenomas are polyclonal [10,83,84].

Clinical observations have prompted more intensive research into the cellular and environmental mechanisms affecting the tumor cell proliferation rate. Proliferative abnormalities of the normal colonic mucosa have been proposed as a possible marker of increased susceptibility to CRC development (particularly an upward shift of the proliferative compartment in the normal mucosa of CRC patients) [85]. Significant differences in the effects of the same therapy (CTx and RT) in patients with the same type of CRC have also been noted [86,87,88]. Attention has been drawn to the predictive (efficacy of different treatment options, individualization of treatment) and prognostic values (treatment outcome) of proliferation rates as a biological CRC feature. The prevailing view was that the pool of rapidly proliferating and mature tumor cells within the tumor was responsible for treatment failure [86]. Increased proliferation rates were considered as one of the determining factors in the accelerated repopulation of malignant tumors including CRC [89,90]. Therefore it was necessary to assess the tumor growth rate as early as possible (i.e., before treatment) to prevent recurrence. Although the prognostic significance of rapid tumor cell proliferation has not been demonstrated, there is a consensus that rapidly proliferating tumors should be treated with accelerated RT regimens. When it comes to CRC, there are huge discrepancies regarding how RT should be administered in rectal cancer (RC). There is no international consensus regarding the preoperative RT irradiation schedule for RC [91].

The main culprits of treatment resistance, metastasis, and relapse in CRC appear to be CSCs [92,93]. These cells are mostly ‘quiescent’ and poorly differentiated and thus can easily survive CTx. The high heterogeneity of TICs was first shown in colon cancer, with only specific subpopulations (self-renewing long-term TICs, LT-TICs) leading to the development of metastatic disease. Other examples of this subgroup include tumor transient amplifying cells (T-TACs) and rare delayed contributing TICs (DC-TICs) [14]. Moreover, abnormal activation of multiple cellular pathways (e.g., Wnt, Notch, Hedgehog, PI3K/AKT) in CRC can result in the emergence of CSCs characterized by excessive self-renewal, increased invasiveness, and resistance to treatment [92].

It appears that varied therapy effects did not occur due to differences in the proliferation rates between CSCs and more differentiated tumor cells as the therapy-induced deaths did not depend on the proliferative status of the cells. These results confirm that CSCs are selectively resistant to conventional CTx due to reduced mitochondrial priming [94]. Studies indicate that these cells arise from normal proliferating colonic crypt SCs. The marker of these cells, encoded by the *LGR5* gene, is overexpressed during CRC development. At the same time, *LGR5* is associated with Wnt pathway activation and the *c-MYC* protooncogene, and may be a prognostic factor in CRC [95].

## 4. Methods to Assess Cell Proliferation in Colorectal Cancer

In clinical practice and basic research, several methods exist for assessing the growth rate of normal and tumor cells. The most common is the assessment of (1) the “density” of ongoing mitoses in the tissue material, known as the mitotic index (the percentage of mitoses in the assessed pool of tumor cells per 1 mm^2^), the so-called mitotic rate, the rate at which cells enter the mitotic phase (M phase) (% of the cells/h) [96,97,98]; (2) the percentage of cells in the S phase by calculating the so-called bromo-, iododeoxyuridine labeling index (LIBrdIUdR) [99,100,101,102] with in vitro tritiated thymidine [103,104], or with a new thymidine analog, 5-ethynyl-2-deoxyuridine (EdU) labeling [82]; (3) IHC expression of classical proliferative markers (e.g., cyclins, proliferating cell nuclear antigen (PCNA), and Ki-67 [105,106,107,108,109]); (4) computed tomography (CT) with dual-layer spectral detector CT [110] or positron emission tomography (PET) [111,112,113].

The cancerogenic process of the colonic mucosa is associated with the development of cell proliferation abnormalities, which precede the onset of morphological alterations such as epithelial dysplasia. Individuals with gastrointestinal (GI) tract cancer risk factors and animals exposed to carcinogens mainly show an increase in the cell proliferation rate and abnormalities in the distribution of proliferating cells. The so-called extension of the proliferative compartment was observed even when the mucosa was not yet affected by morphological abnormalities. This proliferative feature seems to be related to the presence of defects in cell differentiation [114]. There is also a report in which a significantly lower expression of multi-gene proliferation signature (GPS) was observed in CRLMs, confirming lower levels of their proliferation using qRT-PCR and Ki-67 immunostaining. According to the authors, slow proliferation is a biological feature of both CRLMs and primary tumors with metastasis capacity [115]. In the context of the stem cell hypothesis of CRC development, in vitro studies based on the exposure of CSC480 cells to a 2 h pulse of 10 μg EdU have recently allowed for the identification of as many as five different cell populations, of which the EdU-negative and CD44-positive population may represent the ‘true’ CSC lineage [82].

In formalin-fixed, paraffin-embedded tissues, changes in DNA content or the expression of proteins involved in the cell cycle in dysplastic, precancerous, and neoplastic tissues of the human colon were most often comparatively assessed. However, changes in the expression of IHC markers (e.g., PCNA, p53, Ki-67) at different developmental stages of CRC were not always clear enough to serve as reliable prognostic markers [31,116,117].

### 4.1. Assessment of Mitosis in Cancer Tissues

The mitosis count/mitotic index in pathological samples allows for the assessment of tumor proliferative activity, facilitates tumor classification and diagnosis, assesses grade malignancy, determines aggressive behavior, allows for intratumoral lymphocyte counts, and may present prognostic significance [98,118]. The preferred sites for mitosis counting include invasive fronts (rich in viable tumor cells) or the periphery of the tumors. The tissue area for counting mitotic activity for different tumors was standardized as the number of mitoses in a fixed number of high-power fields (HPFs) (typically 10 fields of view at x 400 magnification) [118]. HPFs for digital pathology, different from glass-slide HPFs in conventional light microscopy, require re-evaluation [119]. The current recommendation for CRC is not to report the number of mitoses in HPFs, but to report them per square millimeter [98], or per 2 mm^2^ (this is approximately equivalent to 10 HPFs on modern microscopes) [97,118].

### 4.2. DNA Ploidy and Percentage of Cells in S Phase

Aneuploidy refers to an abnormal number of chromosomes in a cell, different from a multiplication of the haploid set, resulting from several genetic alterations. It reflects both gain/loss of whole chromosomes and unbalanced chromosome rearrangements (e.g., deletions, amplifications, translocations of large genome regions) [120]. For more than 100 years, aneuploidy has been postulated as a tumor-promoting factor, and its clinical relevance is still highlighted as a prognostic marker [121,122]. Interestingly, it has been suggested that tissue SCs have also developed their distinct response to aneuploidy, being able to survive and proliferate as aneuploid [121].

DNA content and ploidy were evaluated as prognostic factors in CRC [123,124,125,126], with DNA aneuploidy demonstrated to be a feature of tumors with a higher proliferation rate [124,126,127]. On the other hand, ploidy alone, determined by flow cytometry (FCM), had no prognostic significance in CRC (DFS). In a group of more than 400 CRC patients, it was shown that nearly 73% of patients showed aneuploid tumors. Still, the DNA pattern was not correlated with either age, gender, location, differentiation, or stage of the tumors [128].

Review studies [129,130] and a meta-analysis [127] indicate a significant association of aneuploidy with tumor progression and a worse prognosis. An older meta-analysis (2007) showed that patients undergoing surgical resection of aneuploid CRC have a higher risk of death after five years [129]. Later meta-analysis (2015) including more than 7000 CRC patients showed a higher prevalence of aneuploidy in late versus early stage sporadic CRC (OD 1.51, 95% CI 1.37–1.67), indicating that genomic instability increases with CRC progression. In 54.1% of studies, a significant effect of aneuploidy on prognosis was described for OS, disease-specific survival (DSS), and recurrence (relapse)-free survival (RFS). Hence, aneuploidy may be considered as a tumor stage-specific prognostic marker [127].

Other methods to assess the proliferation of different cell populations in CRC include evaluating the number of cells in which DNA synthesis occurs using LIBrdIUdR, with tritiated thymidine [103,104] and EdU labeling [82]. Such procedures allow in vivo calculation of the S-phase fraction labeling index (LI), the duration of the S phase (Ts), and the potential tumor doubling time (Tpot) [131].

Evaluation of the binding index of BrdUrd/IdUrd/tritiated thymidine, etc., is possible (1) following the use of monoclonal antibodies (mAbs) against thymidine analogs in FCM or (2) by using the IHC method. Although these method variations are inexpensive and easy to perform, they are characterized by high subjectivity in the evaluation of specimens, poor reproducibility of results, and the lack of standardization between centers [104,124,128,132].

Studies from the 1990s showed that examining only the total and aneuploid LI in CRC is not sufficient as an indicator of proliferation, as Ts also can vary between tumors and even within a single tumor (from 4.0 to 28.6 h). The mean Tpot ranged from 1.7 to 21.4 days. None of the cellular kinetic parameters correlated with Dukes’ classification or histologic examination [133]. Wilson et al. showed that while IUdR assessed by FCM (IUdRfmc) and assessed by IHC (IUdRimm) correlated with each other, and their LIs were significantly higher in aneuploid than diploid tumors, no prognostic property of these markers was demonstrated [124]. Similar results were reported by other authors [126]. On the other hand, Palmqvist et al., using both IUdR detection techniques (FCM + IHC), demonstrated that patients with Dukes’ B tumors with higher IUdR LI (in invasive margin) and/or low Tpot (at both the invasive margin and the luminal border) had longer survival [100]. FCM studies on the prognostic value of the DNA index or S-phase fraction also did not demonstrate prognostic significance for disease recurrence in CRC stages II and III [125], survival in the overall group, or within stages [132]. In contrast, the kinetic parameters assessed by Michel et al. using in vivo injection of Brd and FCM, were independent prognostic factors in diploid tumors. These included lymph node (LN) involvement, ploidy, and Tpot in all tumors, and Tpot only in diploid tumors [131].

In summary, most studies failed to demonstrate the prognostic value of the CRC proliferation markers assessed. Moreover, using these methods, more accurate results for evaluating normal and tumor cell proliferation are obtained after analyzing material at different stages of CRC development. On the other hand, performing such tests before and during treatment allows one to predict the outcome of CRC radiotherapy. For example, BrdUrd LI before RT treatment of RC was not a predictor of early clinical and pathological tumor response. In contrast, the BrdUrd LI ratio before/after RT was correlated with the inhibition of proliferation in responsive tumors. Thus, the rapid growth rate of preoperatively irradiated rectal cancer was a favorable prognostic factor [134].

### 4.3. Immunohistochemical Methods for the Detection of Proliferative Markers

The immunohistochemical (IHC) technique is based on antibodies against specific antigens in tissues and cells. In histopathology, IHC testing is most commonly performed on formalin-fixed, paraffin-embedded tissues that can be stored for long periods of time [135,136]. Increasingly, tissue microarrays (TMAs), which contain selected tissue material from tumors, normal tissues (control), and tumor metastases on a single slide, are being used for IHC. Although the cost of producing TMAs remains high, their selection saves labor time and the number of reagents used (including sometimes expensive antibodies), allowing for better result reproducibility [137]. The markers most commonly used to assess tumor proliferation rate (including CRC) are discussed below.

#### 4.3.1. Thymidylate Synthase (TS) in CRC

Thymidylate synthase (TS, EC 2.1.1.45) is an enzyme protein required to synthesize and repair DNA. It catalyzes the conversion of 2’-deoxyuridine-5-monophosphate (dUMP) to deoxythymidine-5’-monophosphate (dTMP), which is phosphorylated to the triphosphate state (dTTP), a direct precursor for DNA synthesis. It is also an important cellular target for cytotoxic drugs of the fluoropyrimidine group, which are widely used to treat solid tumors [138]. The first clinically used TS inhibitor was the 5-fluorouracil (5-FU) antimetabolite drug, a metabolite of 5-fluorouracil, fluoro-deoxyuridine monophosphate, which forms a ternary complex with TS and 5,10-methylenetetrahydrofolate [139].

IHC studies of TS are used to determine proliferative indices and drug resistance [138]. The role of ectopic production of human TS in the neoplastic transformation of mouse cells in vitro and in vivo has also been demonstrated, suggesting a role for TS as an oncogene [140]. Overexpression of TS is responsible for the resistance of tumor cells to TS-targeted chemotherapeutics and correlates with response to targeted CTx [141,142,143,144]. With the generation of mAbs against TS, particularly TS 106 and TS 109, it was possible to use IHC methods to detect TS in normal and tumor tissues. The color reaction is granular and occurs in the cytoplasm of cells. TS has been shown to be overexpressed in tumors including CRC. The prognostic significance of this IHC marker has also been studied [141,142,143,145,146,147].

Observations on the prognostic role of TS indicate that increased tissue expression of TS may serve as an independent factor of poor prognosis for DFS and OS [141,143,145,146,148,149,150] or RFS and OS [147]. However, there are also results in which the prognostic role of TS in the survival of CRC patients could not be proven. Moreover, it was shown that high levels of Ki-67 were associated with increased (decreased) survival in patients with a low (high) expression of TS [142]. The meta-analysis by Popat et al. showed that tumors with high TS levels appeared to have worse OS compared to tumors with low TS levels (HR 1.74, 95% CI 1.34 to 2.26) [151].

The predictive role of TS in CRC adjuvant therapy has also been investigated in various combinations (e.g., 5-FU-based CTx, oxaliplatin followed by 5-FU). One study showed a significantly higher degree of TS immunoreactivity in primary tumors compared to corresponding metastases. Still, the response rates after CTx for metastatic disease were similar for patients with low and high levels of TS shown in their primary tumors. In contrast, response rates were found to be higher in patients with low versus high TS in metastatic disease (71% and 23%, respectively) [152]. Thus, TS levels in primary tumors cannot be reliably used to predict the response to adjuvant therapy [147,152]. An opinion questioning the benefit of TS labeling for predicting the effect of 5-FU in CRC can also be found in a review paper [139].

Moreover, while an extensive prospective analysis showed that high TS levels in the tumor were associated with improved DFS and OS after adjuvant treatment of CRC, TS expression in the tumor did not predict the benefit of 5-FU-based CTx [153]. However, a recent study by Badary et al. showed that high TS expression is a predictor of early failure in CRC therapy. Hence, high TS expression may help identify patients who will benefit less from oxaliplatin and 5-FU CTx (FOLFOX) [143].

#### 4.3.2. Cyclins in CRC

The human cyclin family includes about 30 genes encoding protein products containing the so-called cyclin box. Only a few subfamilies of these proteins (A-, B-, C-, D-, and E-cyclins) play a role in cell cycle regulation [57,58,61,154]. Others distinguish between ‘primary’ cyclins (A, B1, D1, D3, and E), crucial for cell cycle progression, and ‘secondary’ cyclins (C and H), with indirect cell cycle-related effects. Few papers have addressed the secondary prognostic role of cyclins in cancer including CRC [155].

Cyclin A can activate two different Cdks, playing a role in both the S phase and mitosis (M) [156], controlling various phenomena related to DNA replication and progression through the G2 phase [58]. Cyclin B is a regulator of the mitotic phase, responsible for M phase entry and chromosome segregation. In turn, cyclin C, encoded by the *CCNC* gene, is involved in G1/S progression. It forms complexes with cdk8 and cdk19, modulating DNA initiation and duplication by binding Mdm2 binding protein (MTBP), an interaction required for proper entry into the M phase with complete DNA replication [157]. D-type cyclins are a major determinant of cell cycle initiation and progression in many cell types. Cyclins D1, D2, and D3 (encoded by *CCND1*, *CCND2*, and *CCND3*) are identified as cell type-specific G1 mitogen sensors. The E-type cyclins control DNA replication. Cyclin E1, encoded by the human *CCNE1* gene, interacts mainly with Cdk1 and Cdk2 and plays an essential role in transition of human cells from G1 to the S phase [58,158].

In some studies, cyclin A (A2) overexpression was observed in 77–80% of CRC cases [159,160]. In rectal cancer, a linear correlation was observed between cyclin A and Ki-67-positive cell expression, whereas no such relationship was found between TS and cyclin A [161]. Several publications have recognized cyclin A overexpression as an independent unfavorable prognostic factor in CRC patients [159,160,162]. There are also reports showing that high cyclin A expression was independently associated with improved survival [155], and its level above the median predicted a better prognosis in CRC patients (HR 0.71, 95% CI 0.53–0.95) [163].

Cyclin B (B1) is classified as a mitotic cyclin [164,165]. Its elevated expression may promote the development of CRC, but its prognostic significance is controversial. Decreased expression of this cyclin has been shown in pCRC cases characterized by large size, mucinous type, deep invasion, or short postoperative survival. High cyclin B1 expression has been associated with increased p53 levels in ADs, and high Ki-67 in ADs and primary carcinomas [164]. Cyclin B1 is overexpressed and promotes cell proliferation in early-stage CRC [165,166]. No correlation was found between cyclin B1 expression and DFS or OS [165]. Other authors have reported that after CRC cells invade surrounding tissues and metastasize to distant tissues, cyclin B1 expression is reduced. Furthermore, it was observed that patients with a low level of cyclin B1 had lower survival rates than those with a high level of cyclin B1 expression. Suppression of cyclin B1 may promote tumor cell migration and invasion and reduce E-cadherin expression. Cyclin B1 may thus promote tumor growth but inhibit metastasis in CRC [166]. As shown in a recent meta-analysis regarding the prognostic role of cyclin B1 in solid tumors, in CRC, elevated cyclin B1 expression was associated with better prognosis, reflected by favorable 5-year OS of CRC (OR 0.49, 95% CI 0.30–0.82) [167].

Cyclin C overexpression was observed in 88% of CRCs, and *CCNC* (qRT-PCR) amplification was independently associated with poor prognosis. The association between *CCNC* amplification and impaired survival appears independent of its gene product [155]. However, further studies on the role of cyclin C itself as a prognostic factor in CRC are lacking. In contrast, a study by Firestein et al. found the expression of Cdk8, a kinase functionally related to cyclin C, in 70% of CRCs. This expression was independently associated with β-catenin activation, female gender, and fatty acid synthase (FASN) overexpression. Cdk8 expression also significantly increased the colon cancer-related mortality. However, no such association was observed among RC patients. These data support a potential association between Cdk8 and β-catenin and suggest that CDK8 may identify a subgroup of CRC patients with poor prognoses [168].

D-type cyclins play a central role in cell cycle entry. Changes in the activity of the D-Cdk4/6 cyclin complex are an almost universal feature of cancer cells [60]. Their expression increases in response to oncogenic alterations in key oncogenic pathways (e.g., K-RAS, PI3K/AKT, WNT) [58]. Overexpression of cyclin D1 is observed in CRC, particularly in advanced disease [160,169,170,171]. However, opinions are divided on the prognostic significance of this cyclin. Overall, more than 20 publications have been published on the prognostic value of cyclin D1 expression in case–control studies, as reviewed by other authors [171]. Maeda et al. showed a shortening of both OS and DFS, and an increase in the CRC recurrence rate in patients with strong cyclin D1 expression [172]. In the study by Bahnassy et al., as mentioned, cyclin D1 overexpression in CRC, similarly to cyclin A, was also correlated with shorter OS. This study indicated that cyclin D1 amplification was also associated with reduced OS. Both cyclin D1 and cyclin A were independent prognostic factors in CRC patients [160]. Another study showed an association between increased cyclin D2 and D3 expression and vascular invasion, CRLM, and decreased DSS [173]. In turn, a study by Mao et al. showed that positive cyclin D1 expression was associated with shorter survival in patients with colon adenocarcinoma [174]. Another publication demonstrated worse 5-year survival in patients with positive cyclin D1 expression in advanced-stage CRC (III, IV) [169]. Moreover, a recent study showed that cyclin D1 and epidermal growth factor receptor (EGFR) overexpression and late pathological stage after surgery were characterized by shorter relapse-free time (RFT) [175]. It has also been shown that the early recurrence of CRC in high-risk Duke B and Duke C stages is associated with high cyclin D1 expression [176]. However, some studies reported no prognostic role for cyclin D1 in RC or CC [155,177,178,179,180]. Finally, some studies have considered cyclin D1 overexpression to be a good predictor of survival [181,182], both in terms of cytoplasmic and nuclear expression [183].

There are also two meta-analyses on the prognostic significance of cyclin D1. One of them (2014) showed that cyclin D1 overexpression is a factor for poor prognosis in CRC, both in terms of OS (HR 0.73, 95% CI 0.63–0.85) and DFS (HR 0.60, 95% CI 0.44–0.82) [170]. Another meta-analysis (2022) confirmed these results, reporting both shorter OS (HR 0.36, 95% CI 0.94–0.22) and DFS (HR 0.46, 95% CI 0.77–0.20) [184].

In contrast, in a study by Jun et al. based on a large cohort of pCRC patients (n = 495), in which high cyclin D1 expression was observed in nearly 80% of patients, high cyclin D1 expression was a marker for better OS and RFS. Multivariate analysis showed that cyclin D1 overexpression and the young age of patients remained independent predictors of higher OS rate. In turn, high cyclin D1, female gender, CTx, absence of nodal metastasis, and lower T category remained independent predictors of better RFS. The authors believe that cyclin D1 expression can be a favorable prognostic indicator in CRC patients [171].

Studies on the prognostic role of cyclin E in CRC have also been conducted, most often together with other markers of cellular proliferation (e.g., cyclin D1 and Ki-67) [176,179,185,186]. Ioachim et al. demonstrated cyclin E overexpression in 30% of CRC patients, but the prognostic significance in determining the risk of recurrence and OS was not confirmed [179]. Elevated cyclin E expression correlated with increasing TNM staging and decreasing tumor differentiation. In turn, PFS and median survival were reduced in patients with positive cyclin E expression [185]. Another group found cyclin E expression in a similar proportion of CRC patients (~35%) but did not report prognostic significance in CRC as a single marker [186].

The data in Table 1, arranged chronologically, show variable results regarding the tissue expression of various cyclins in CRC. In general, the overexpression of these cell cycle markers is detected in most patients (up to almost 90%). When it comes to the evaluation of the prognostic value of cyclins, most data concern cyclins A (A2), B (B1), and D (D1). However, as with other tissue markers, the data are not consistent. Cyclins of the D family show the strongest association with signaling pathways involved in CRC development (e.g., KRAS, PI3K/AKT, WNT). Moreover, while some studies have also associated cyclin D1 overexpression with poor prognosis, some present results describe no prognostic significance or indicate cyclin D1 as a good prognostic factor. In conclusion, examining the expression of these proteins alone seems insufficient to determine the prognosis for survival of CRC patients. Hence, further research is needed to determine the role of cyclin C and E as prognostic markers in CRC.

#### 4.3.3. Proliferating Cell Nuclear Antigen (PCNA) in CRC

In eukaryotic cell physiology, PCNA plays a vital role in DNA replication and many replication-associated processes. It is a 36 kDa non-histone nuclear protein, accompanying delta and epsilon DNA polymerase. It is referred to as a cyclin, playing a prominent role in cell proliferation. It is mainly produced in proliferating and transformed cells as a specific marker of cell division [187]. PCNA expression is detected in all phases of the cell cycle, confirming the function of this polypeptide in DNA repair, synthesis, and regulation [108].

There is a significant variability of results regarding PCNA expression in ‘adenoma–carcinoma sequence’ changes in CRC. Either no increase in PCNA-positive cells was detected in adenocarcinoma [116], or a gradual increase in PCNA expression was shown in HP-AC lesion sequences [117]. Some authors observed high PCNA expression in more aggressive forms of ADs, which can progress to malignant lesions [188]. As for the value of PCNA expression in predicting CRC, results also vary. One publication recognized PCNA as an independent predictor of relapse and shorter survival in CRC patients [189]. Choi et al. demonstrated a significantly higher relapse rate in CRC patients, with higher-than-average PCNA-LI. Also, the four-year survival rates in cases with higher-than-average PCNA-LI were considerably worse than those with lower-than-average PCNA-LI [190]. Other studies either failed to demonstrate the prognostic value of PCNA in this cancer [123,124,191] or showed an inverse relationship between the percentage of PCNA-positive cells and the survival time of CRC patients [192]. Increased PCNA-LI of tumors was often associated with tumor progression (venous invasion, lymph node metastasis, or liver metastasis), while higher PCNA-LI was also associated with less differentiated tumors. Thus, PCNA testing could have prognostic significance for assessing higher malignant potential [193]. However, Neoptolemos et al., in RC studies, showed that PCNA-LI was not prognostic in this cancer subtype and that patients with the smallest LI exhibited the worst survival times [191]. In contrast, Nakamura et al. showed longer survival for CRC patients with lower PCNA expression, which was true for both CEA-positive and serum CEA-negative patients [194]. Some authors have indicated that while higher proliferation is associated with a higher incidence of rectal ADs, PCNA-LI is not useful for predicting future colorectal neoplasia [195]. Others have found lower PCNA-LI expression to be a good predictor of survival, especially in combination with HLA-DR expression [196]. Studies of the entire cell cycle panel (e.g., cyclins D1, E, cyclin-dependent kinase (CDK) inhibitors: p21 and p27) and other cell cycle regulators including PCNA have not proven the prognostic value of any of them in terms of predicting the risk of relapse or OS. These molecules have mainly been considered as cell growth regulators during colorectal carcinogenesis [179]. Moreover, studies by Guzinska et al. confirmed correlations between PCNA expression and lymph node metastasis and tumor location (lower in RC). However, the prognostic value of PCNA was not evaluated [197].

The only available meta-analysis on the level of immunohistochemical PCNA expression as a prognostic factor in CRC considered OS, cancer-specific survival (CSS), and DFS in 1372 CRC patients [198]. It showed that patients with high PCNA expression were characterized by shorter OS (HR 1.81, 95% CI 1.51–2.17) and CSS (HR 1.99, 95% CI 1.04–3.79). However, there was no significant association between PCNA and DFS. Thus, it was shown that high PCNA expression can predict a poor prognosis in CRC patients. However, this analysis needs to be confirmed in a larger number of studies based on bigger groups of patients.

Table 2 shows, in chronological order, the results that illustrate the difficulty in forming a clear opinion on the prognostic significance of PCNA. The tissue expression of PCNA was studied simultaneously with various histopathological classifications and/or with other tumor biomarkers (e.g., CEA, HLA-DR, Bcl-2) to analyze the interactions of these proteins and/or to expand the panel of prognostic factors in CRC.

#### 4.3.4. Ki-67 Antigen in CRC

The prototype of the Ki-67 antigen was the IgG1 class, murine mAb, directed against the nuclear fraction of the Hodgkin’s lymphoma-derived cells (L428) [106,199]. Recognition of the structure of the Ki-67 protein (pKi-67) has enabled this protein to be placed in a new category of cell cycle-related, nuclear non-histone proteins. pKi-67 is encoded by the *MKI67* gene on chromosome 11 (10q26) and has two major splice variants of 320 and 352 kDa [200]. The pioneering generation of anti-Ki-67 mAbs [199], characterization of the Ki-67 antigen using molecular biology techniques [201] and experimental studies demonstrated the presence of pKi-67 in the S, G2, and M phases of the cell cycle, and its absence in the G0 phase [105]. Thus, Ki-67 exhibits the so-called growth fraction in non-cancerous cells and tumors, indicating it as a marker of cellular proliferation [105,202]. It is worth mentioning that Ki-67 positivity does not always indicate that the cell entered the division phase. It can also signify a transition into a quiescent state and the possibility of entering the cell cycle after removing the inhibiting factor. Subcellular localization during interphase shows the presence of Ki-67 mainly in the cell nucleus, while in mitosis, it is translocated to the surface of chromosomes [106]. Recent studies have also indicated the extranuclear translocation of pKi-67 in non-cancerous cells to eliminate the protein, with initial accumulation in the endoplasmic reticulum (ER) and later in the Golgi apparatus. This mechanism is less effective in cancer cells [203].

Numerous publications over the years have discussed the role of Ki-67 as a marker of proliferation [106,202,204,205,206] and thus as a prognostic factor in many diseases, primarily cancer (including CRC) [207]. At the same time, studies have been conducted on the structure and biological role of pKi-67 in normal cells [207,208,209,210]. The multifactorial regulation of Ki-67 in non-cancerous and cancerous human cells has been described [203,206,210]. The role of Ki-67 in cell cycle progression is debated, most notably its almost opposite role in the initial phase of mitosis (prometaphase) (chromosome individualization) and exit from mitosis (chromosome clustering) [211]. Ki-67 has been shown to form repulsive molecular brushes during the early stages of mitosis [212]. In turn, the brushes collapse during mitotic exit, and Ki-67 promotes chromosome clustering [213]. Other significant advancements regarding the structure and functional role of pKi-67 in recent years include the demonstration of (1) the putative role of this protein in the higher-order organization of perinucleolar chromatin [208]; (2) the involvement of pKi-67 in the early stages of rRNA synthesis in vivo [209]; (3) the involvement of Ki-67 as a PP1 interacting protein (PIP) in the phosphorylation of nucleophosmin/B23 by casein kinase II (CKII) and the organization of the perichromosomal layer [214]; (4) the role in the generation of a spherical and electrostatic charge barrier, enabling independent chromosome mobility and efficient interaction with the mitotic spindle [212]; (5) the role in the spatial organization of heterochromatin in proliferating cells and in the control of gene expression [215]; (6) the differential regulation of the two main splice variants of the protein (i.e., α and β) in non-cancerous and cancerous cells; (7) the continuous regulation and degradation of Ki-67 by proteasomes in normal and cancerous cells and the extranuclear pathway of protein elimination [203]; (8) changes in expression depending on cell cycle regulation as a reliable indicator of the effect of CDK4/CDK6 inhibitors on cell proliferation [109]; (9) accumulation of the protein during the S, G2, and M phases, and degradation during the G1 and G0 phases; (10) the graded, rather than binary, nature of the protein, with a stable decrease in pKi-67 levels in quiescent cells [210]; (11) the presence of a gradient of Ki-67 expression depending on the phase of the cell cycle, (fast-growing tumors exhibit high levels of this protein in G2 phase cells, while in slow-growing tumors, these levels are notably lower) [216]; (12) the involvement in the regulation of chromosome clustering conditioning the removal of mature ribosomes from the nucleus after mitosis [213]. Although Ki-67 is widely recognized as a proliferation marker, genetic studies indicate that its levels do not correlate directly with this process. Indeed, the downregulation of Ki-67 did not affect the proliferation of HeLa cells [212,214,215], BJ-hTERT cells, and U2OS cells [215].

To evaluate the Ki-67-positive cells (the Ki-67 labeling/proliferating index, LI, PI) in paraffin-embedded sections during histopathology, an antibody called MIB-1 is most commonly used. Sometimes, in the literature, the name pKi-67 is used interchangeably with anti-Ki-67 antibody, or both are used together (Ki-67/MIB-1). Using IHC, it is possible to not only determine the presence of Ki-67 LI but also identify the type of proliferation, which could be a potential prognostic factor [70].

##### Ki-67 and Clinicopathologic Data in CRC Patients

The study of the tissue expression of Ki-67, as the most common proliferative marker, is widely used to assess tumor grade or stage, predict tumor progression, or identify potential therapeutic targets [106,202,204].

Lower Ki-67 LI with medium intensity has been described in non-neoplastic polyps compared to neoplastic lesions [217]. In colorectal ADs, the Ki-67 expression was lower [218] or comparable to CRC [217]. Moreover, a higher positive rate was observed in AD cases with high atypia and carcinoma in situ [219] and in more severe dysplastic adenomatous lesions [220]. The latter study indicated that the severity of dysplasia is associated with greater cellular proliferation, as opposed to the morphological type of AD (tubular, tubulovillous, and villous).

High levels of Ki-67 expression in pCRC are most often correlated with more severe histopathological changes (stage, grade) [217,218,221,222,223,224,225,226,227,228,229,230]. An inverse correlation between Ki-67 expression and the degree of differentiation in non-mucinous AC was observed [231]. A higher Ki-67 LI (≥30%) was present in lymphatic and venous invasion as well as in lymph nodes and CRLMs. The same LI (≥30%) in the primary tumor was associated with a significantly higher incidence of metachronous CRLMs. However, the mean Ki-67 LI was higher in primary tumors compared to CRLMs [221], which was also confirmed at the mRNA level [115]. Moreover, other researchers observed a positive correlation of Ki-67 LI with LN metastasis [88,197,224,228,229,232]. At the same time, Lei et al. showed Ki-67 level ≥ 60% to be associated with a high risk of distant metastasis and death, compared with a Ki-67 below this level [230].

Some authors did not show any significant correlation with the clinicopathological data [96,124,126,142,233,234,235]. In contrast, other authors have shown better clinicopathological variables in CRC patients with higher Ki-67 expression [147,236,237] and an inverse relationship between Ki-67 expression and tumor aggressiveness [115]. Similarly, the percentage of Ki-67-positive cells in poorly differentiated and mucinous AC was significantly lower than in well-differentiated and moderately differentiated AC. In contrast, lower Ki-67 LI in the primary lesion in cases with metachronous liver or lung metastases, compared to synchronous cases, may indicate that metachronous hematogenous metastases occur even in tumors with low proliferative activity [219].

In numerous publications, IHC studies have also evaluated other proliferation markers and their correlation with Ki-67 expression levels. IudR [124] and BrdUrd [126] were positively correlated with Ki-67, while TS expression correlated with Ki-67 in one study of RC [161] but not in others [142,238].

Positive correlations with Ki-67 were observed for cyclin A [141,159,160], cyclin B1 [164], cyclin D1 [160,169], cyclin E [185], cyclin E, and the p21waf1/cip1 cdk inhibitor [179]. Many authors have investigated the extent of cellular proliferation, measured by the expression of Ki-67 and the mutated tumor suppressor gene product p53, as an example of the most common genetic aberration in CRC [125,142,147,218,224,228,232,235,237,239,240,241,242]. As for the reciprocal correlations of the two proteins, either a directly proportional correlation [228,235,241], no significant correlation [224,238], or an inverse correlation of Ki-67 and p53 was detected [218].

##### Ki-67 as a Prognostic Marker in CRC

Cell proliferation is significantly associated with CRC progression and can be used to identify patients with a predicted unfavorable disease outcome after surgery [223,243]. The prognosis of CRC is not solely determined by the proliferative capacity of tumor cells [224]. Many clinicopathological prognostic factors have been documented, related to the advanced pathological TNM stage (pTNM) and the so-called TNM-independent factors (e.g., tumor subtype and histological grade, lymphovascular invasion, tumor-infiltrating lymphocytes, perineural invasion, microvessel density, tumor margin configuration, and poorly differentiated clusters (PDCs) [55,244,245,246]. One publication provided an algorithm to profile ‘bad’ and ‘good’ prognostic biomarkers in CRC that considered the clinical features, histopathology, biochemical markers, and response factors. Of those discussed in this review, typical proliferative markers and, at the same time, unfavorable prognosis factors, included cyclin D, TS, and PCNA [244,245]. Another review reported that more than 100 differentially expressed CRC molecular markers (including proliferative markers), representing more than 1000 biological pathways, have been demonstrated in CRC [55]. It should also be mentioned that MSI-H status and impaired signaling pathways resulting from common gene mutations in CRC (e.g., *WNT*, *TP53*, *KRAS*, *BRAF*, *PI3K*, *TGF-β*, phosphatase and tensin homolog protein (*PTEN*)) or amplifications of specific genes (e.g., *IGF-2*, *IGFBP2*, *EGFR, VEGF*, *SMAD*) are usually associated with the overexpression of markers and lead to increased cell proliferation and the inhibition of apoptosis [55,244,245].

Many studies from different regions around the world have also shown the importance of the tissue overexpression of Ki-67 in pCRC and/or CRLM as a poor prognostic predictor of survival for patients with this cancer. Most publications have shown that high Ki-67 expression was associated with inferior OS, but some reports have demonstrated that high Ki-67 expression was correlated with favorable/longer survival [237,247,248,249,250], also in CRLM [238]. There was also a study in which the Ki-67 LI analysis results demonstrated various proliferation extents in the central areas of the tumor (cPDCs) (high) and at the tumor periphery (pPDCs) (low) and a range of different correlations with the clinical data [246].

It should be noted that few publications have investigated the prognostic significance of Ki-67 expression in different CRC locations (colon/rectum), resulting in divided opinions. One research group reported no correlation between Ki-67 expression, tumor location, and prognosis [237]. In contrast, Hilska et al. demonstrated a better prognosis for Ki-67 LI values ≥ 5% compared to a lower index, only in the group of patients with rectal cancer [182].

Several authors have indicated Ki-67 as an independent prognostic factor. For some, an increase in Ki-67 is a poor prognostic factor for survival [96,223,228], while others have reported a longer survival in patients with high Ki-67 levels [237,250]. An analysis by Valera et al. showed that tumor Ki-67 PI was an independent prognostic variable, consistently used by the classification and regression tree (CART) algorithm to classify patients with similar clinical features and survival [243]. Studies on Ki-67 expression in CRLM indicate the overexpression of this protein as an independent factor of poor OS prognosis [238,251,252].

The meta-analysis by Luo et al., focused on Ki-67 validation using IHC expression, covering 34 studies based on 6180 primary CRC patients, confirmed that the high expression of Ki-67 is a poor predictor for OS (HR 1.54, 95% CI 1.17–2.02) and DFS (HR 1.43, 95% CI 1.12–1.83) based on an univariate analysis. In multivariate analysis after adjusting for other prognostic factors, an association was shown only for OS (HR 1.50, 95% CI 1.02–2.22) [175]. Another meta-analysis investigated the determination of prognostic biomarkers in CRLM. Ki-67 was included among the 26 independent OS biomarkers in resected CRLM [253].

More than a dozen research publications on Ki-67 as a prognostic factor in CRC have also investigated the prognostic significance of potential apoptosis proteins (e.g., p53, bcl-2, programmed death ligand 1 (PD-L1), survivin) [125,134,142,147,182,197,218,224,228,232,235,237,238,241,254,255] (Table 3).

There is also a summary of studies on the segmental distribution of some commonly used molecular markers (including proliferative and apoptotic markers) in CRC, which could also potentially affect their prognostic or predictive value [256]. One such marker is Ki-67, a component of the 12-gene Oncotype DX^®^ Colon Cancer Assay, with potential significance for predicting the risk of disease recurrence, DFS, and OS in stages II and III CRC [257]. However, more recent studies indicate that routine use of the Oncotype DX Colon Recurrence Score in stage IIa CC may be unnecessary, especially in patients with normal levels of additional biomarkers [258].

Table 3 shows the potential correlations between Ki-67 expression, clinicopathological data, and survival as prognostic factors in CRC. Moreover, it illustrates the broad geographical coverage of the studies conducted, which include several countries in America, Europe, and Asia. The studies mainly used mAbs (MIB-1) rather than polyclonal antibodies, allowing for better result comparability. However, the publications varied in the semi-quantitative methods used to estimate the results, which may be one reason for the differences between the investigators. Most articles, revealing significant correlations between Ki-67 expression and clinicopathological data, also provided answers regarding the prognosis and survival of patients (OS, DFS).

Traditionally, pathologists examine the expression of IHC markers visually and calculate it semi-quantitatively by considering the intensity and distribution of specific staining. Visual assessment is fraught with problems due to the subjectivity of interpretation. There is a lack of standardized systems for evaluating performance, relying on different cut-off values and inconsistent criteria to define the threshold value of marker/antigen positive expression by IHC. A lower reproducibility of results may also be affected by differences in the preparation conditions, antibodies used, their dilutions, and IHC reaction detection systems [135,136,198,210]. Automated IHC measurements promise to overcome these limitations. Nowadays, spatial visualization methods of digital images are used to quantify IHC data [262].

### 4.4. Modern Molecular Biology Techniques for the Assessment of Proliferative Markers in CRC

With the rapid development of complex molecular biology techniques (e.g., qRT-PCR, in situ hybridization (ISH), RNA/DNA sequencing, NGS, and DNA methylation detection methods), there is a constant search for new biomarkers of cellular proliferation with potential diagnostic, prognostic, and/or predictive significance in cancers including CRC [115,236,259,263,264,265,266,267,268,269,270,271,272,273,274,275,276,277,278].

Quantitative RT-PCR is generally used as the ‘gold standard’ method to measure RNA expression [115,259,267,276,277]. In situ hybridization is a research tool to detect protein production and provides invaluable information regarding the localization of gene expression in heterogeneous tissues. For example, it was used to detect Ki-67 mRNA in CRC tissues with the digoxigenin-labelled cRNA probe [236].

RNA sequencing is used to study the expression of non-coding RNAs (ncRNAs) [275,276], often complementary to methods for assessing protein expression (e.g., IHC, BrdU staining, Western blotting, qRT-PCR, and ISH). Among the sequencing techniques, NGS is currently the only method that enables the parallel sequencing of thousands of short DNA sequences in a single assay, replacing many less advanced profiling technologies. NGS is used to analyze the genome (whole and partial genome), methylome, transcriptome, or available chromatin using techniques including DNA-Seq, RNA-Seq, or chromatin profiling with methods such as ChIP-Seq. This technology offers a better approach for detecting multiple genetic changes with a minimal amount of DNA. What is particularly important is that it is also possible to sequence RNA transcripts from single cells (scRNA-Seq) [46].

Detection methods for DNA methylation in CRC include methylation-specific polymerase chain reaction (MSR), DNA sequencing (e.g., bisulfide sequencing, pyrosequencing), methylation-specific high resolution melting curve analysis (MS-HRM), and MethyLight assay (reviewed in [278]).

#### 4.4.1. PCNA mRNA Expression

PCNA expression was also studied at the RNA level. Yue et al., using RT-PCR, showed higher PCNA mRNA expression (94.1%) in patients with CRC and venous invasion and LM than in CRC without metastasis (70.6%), confirming the increased production of this marker with CRC progression [263]. However, PCNA was not indicated as a prognostic marker but only as a useful marker for evaluating the LM of cancer cells. In contrast, Cui et al., using qRT-PCR, demonstrated increased PCNA antisense RNA1 (PCNA-AS1) expression in CRC relative to the controls, and detected correlations of this biomarker with the clinical data (tumor invasion and TNM stage). A higher expression of PCNA-AS1 was also confirmed by in vitro studies. These data suggest a role for PCNA-AS1 mainly as a diagnostic rather than a prognostic marker in CRC [264].

#### 4.4.2. Ki-67 mRNA Expression

Possible correlations between the Ki-67 mRNA and clinicopathological data were also analyzed, investigating its prognostic significance in CRC [115,236,259]. A positive correlation was described between protein LI, Ki-67 mRNA, and TNM. The mRNA level was also prognostically important as it correlated with patient survival, similarly to the pKi-67 index [259]. The correlation between pKi-67 LI (median: 59%) and the mRNA level detected using ISH (median: 42%) was slightly more difficult to obtain as a positive correlation was observed in 32/47 resected tumors, with a significant difference detected in 15 cases. In the latter tumors, more than 30% of the cells were pKi-67-positive but did not exhibit the presence of its mRNA. The authors explain this by the likelihood of cell cycle arrest. Interestingly, the latter patients were characterized by a better prognosis. In other words, tumors with high pKi-67 and low mRNA are likely to proliferate more slowly and, hence, be attributed to a better prognosis [236]. On the other hand, comparative studies between pCRC and CRLMs, using qRT-PCR and IHC, showed significantly lower multi-gene proliferation signature (GPS) expression in CRLM and confirmed their lower proliferation rate. Interestingly, proliferative activity was significantly lower for primary cancers with recurrence or those with established metastases than for CRCs that did not metastasize and had no recurrences [115]. Such studies need to be continued, as they may shed new light on tumor proliferation.

#### 4.4.3. Non-Coding RNAs (ncRNAs) Expression

Particular value is attributed to fragments of the human genome that do not encode proteins but play a specific role in many of the biological processes involved in colon carcinogenesis including cell cycle regulation. These are the so-called non-coding RNAs (ncRNAs), among which there are two main classes: small non-coding RNAs with less than 200 nucleotides (nc) (e.g., microRNAs, small interfering RNAs, Piwi-interacting RNAs, small nuclear RNAs, and small circular RNAs) and long non-coding RNAs (lncRNAs) (greater than 200 nc in length [270,273,274].

Studies have consistently demonstrated that the majority of both miRNAs and lncRNAs are dysregulated in CRC. The role of hundreds of different ncRNAs has been demonstrated in CRC cell proliferation in vivo and in vitro. Non-coding RNAs most often show increased expression in CRC compared to the controls. Depending on what function a given ncRNA has in the tumor (oncogene, tumor suppressor), its overexpression or downregulation enhances proliferative activity and tumor progression [266,268,270,279,280,281].

Numerous reviews have illustrated the underlying mechanisms of the biological action of miRNAs in CRC and/or reported downstream targets linked to known signaling pathways in colorectal carcinogenesis (mostly responsible for cell proliferation) [276,279,282,283]. For example, microRNAs can activate the KRAS pathway (downregulation of tumor suppressors: miR-96-5b, miR-384, mi-143, Let-7) [279,283] as well as WNT (miR-135, miR-145, miR-17-92) and EGFR signaling (miR-126, miR-143, miR-18a, Let-7, miR-196a, miR-21), and inactivate the TGF-β pathway (miR-200c) [282]. They can result in the downregulation of the TP53 pathway (overexpression of miR-34a, miR-34b, miR-34c, miR-192, miR-194-2, miR-215), epithelial–mesenchymal transition (EMT) (overexpression of miR-181a, miR-17-5p, miR-494; miR-21, miR-22) and SMAD4 (overexpression of miR-20a, miR20a-5p, miR-888). Moreover, the miR-21, miR-31, and miR-200 families are involved in EMT regulation [282,283].

In addition to miRNAs, lncRNAs are also closely involved in enhancing cellular proliferation, acting through the CRC’s well-known signaling pathways, as already described in some excellent reviews [268,281,284,285]. These include (i) JAK/STAT (downregulation of cancer susceptibility candidate 2, CASC2), (ii) MAPK (overexpression of H19 imprinted maternally expressed transcript, H19 and a newly discovered lncRNA with a length of 2685 nc, i.e., LINC00858), (iii) EGFR/MAPK (overexpression of solute carrier organic anion transporter family member 4A1-antisense RNA 1, SLCO4A1-AS1), (iv) Ras/MAPK (overexpression of colorectal neoplasia differentially expressed, CRNDE), and (v) AKT (overexpression of nuclear-enriched abundant transcript 1, NEAT1). A further example would be WNT-β-catenin signaling, which is activated by the overexpression of lncRNAs including small nucleolar RNA host gene 1 (SNHG1), HOX transcript antisense RNA (HOTAIR), SLCO4A1-AS1, taurine upregulated gene 1 (TUG1), and the downregulation of growth arrest specific 5 (GAS5). In turn, the TGF-β1 pathway is affected by the downregulation of maternally expressed 3 (MEG3) and the upregulation of LINC00858, whereas TGF-β/Smad is activated by the upregulation of SNHG6. Many lncRNAs are involved in the regulation of the EMT process. These mainly include TUG1, sprouty RTK signaling antagonist 4 intronic transcript 1 (SPRY4-IT1), and promoter of CDKN1A antisense DNA damage activated RNA (PANDAR) [281,284,286,287]. The activation of proliferation through STAT3 or β-catenin-mediated signaling pathways is also mediated by the upregulation of lncRNAs such as BC200, CASC15, colon cancer-associated transcript 2 (CCAT2), focally amplified lncRNA on chromosome 1 (FAL1), SNHG1, and SnaR. The ERK (MAPK)/JNK pathway is also affected by lncRNA DMTF1V4. Moreover, lncRNA SNHG7 acts in the K-RAS/ERK (MAPK)/cyclin D1 pathway (reviewed in [268]), while the MIR22 host gene (MIR22HG) is responsible for blocking the SMAD complex, resulting in the inhibition of EMT signaling [270,288].

Some of the lncRNAs above-mentioned interact with other cell cycle markers. For example, the zinc finger NFXT-type containing 1 antisense RNA 1 (ZFAS1) affects cell proliferation through a mechanism that destabilizes p53 via the CDK1/cyclin B1 complex [289]. Another lncRNA (i.e., ENSG00000254615), inhibits CRC cell proliferation and attenuates CRC resistance to 5-FU by regulating p21 and cyclin D1 expression [290]. Cyclin D1 also belongs to one of the target proteins of lncRNAs such as SNHG1 [291], SNHG7 [292], and XIST [293]. PCNA, on the other hand, is one of the target proteins for the lncRNA FAL1 [294]. These studies suggest a complex network of functional relationships between ncRNAs and classical cell cycle proteins, which may result in their variable expression at different stages of CRC development. In turn, any epigenetic modifications and interactions of lncRNAs with both miRNAs and proteins as well as the action of lncRNAs as precursors or pseudogenes of miRNAs may regulate the expression of multiple genes [272].

The prognostic role of ncRNAs in CRC has also been increasingly demonstrated. A summary of the activity of both subtypes of ncRNAs (miRNAs and lncRNAs) as regulatory and prognostic factors in CRC is provided in other reviews [266,279,280,295].

##### MicroRNA (miRNAs, miRs)

There has been a rapidly increasing number of original publications and systematic reviews [276,283,296] addressing the prognostic role of miRNAs in CRC. A worse prognosis for survival (worse OS/DFS) is related to both miRNAs that are downregulated and overexpressed in CRC [276,283]. Representative miRNAs detected in tissues and body fluids are often compared in the literature [276]. An analysis of 115 articles identified hundreds of miRNAs with oncogene properties including miR-21, miR-181a, miR-182, miR-183, mi-R210, and miR-224. Overexpression of these miRNAs was associated with CRC progression and shorter patient survival. The most frequently described tumor suppressors among miRNAs included miR-126, miR-199b, and miR-22. Decreased expression of the latter was also associated with poor prognosis and a higher risk of relapse (worse DFS) [283]. A detailed review addresses the mechanisms of methylation of miRNAs as a cause of their silencing and the prognostic value of such altered miRNAs in CRC [296].

In addition, dozens of meta-analyses are available on the prognostic role of single miRNA types (e.g., miR-21 [297], miR-181a/b [298], miR-20a [299], miR-155 [300]) or the entire group of miRNAs tested (e.g., miR-21, miR-215, miR-143-5p, miR-106a, and miR-145) in specific stages of CRC development [301]. Gao et al., in their meta-analysis, showed that the strongest markers of poor prognosis included high levels of miR-141 in blood (HR 2.52, 95% CI 1.68–3.77) and miR-224 in tissue (HR 2.12, 95% CI 1.04–4.34) [302].

##### Long Non-Coding RNAs (LncRNAs)

Modern molecular techniques and the TCGA dataset allow for the identification of an increasing number of different lncRNA subtypes as new prognostic and predictive factors in CRC [267,272]. Representative lncRNAs detected in CRC tissues and plasma/serum (circulating lncRNAs) have already been compared in the literature [303]. A prognostic role was shown for lncRNAs with tumor suppressor and oncogene properties. For example, the reduced expression of tumor suppressors such as LOC285194 [304] or MIR22HG [288] is associated with poor prognosis. Upregulation of lncRNA-oncogenes in CRC has also been associated with poor prognosis through various mechanisms. These include, among others, plasmacytoma variant translocation 1 (PVT1) [275,305], differentiation antagonizing non-protein coding RNA (DANCR) [306], HOXA distal transcript antisense RNA (HOTTIP) [307], BRAF-activated non-protein coding RNA (BANCR) [308], SPRY4-IT1 [309], CCAT1/CCAT2 [310], and X inactive specific transcript (XIST) [311].

Although many ncRNAs have been reported as proliferative markers, only a few meta-analyses have provided evidence for the actual role of selected lncRNAs in CRC prognosis [265,312,313,314,315,316]. These include, among others, overexpressed oncogene urothelial cancer-associated 1 (UCA1) for OS (HR 2.25, 95% CI 1.77–2.87) [312], or SNHG6 for OS (HR 1.92, 95% Cl 1.48–2.49), and DFS (HR 1.84, 95% CI 1.02–3.34) [313]. As shown in a recent meta-analysis based on 25 publications and more than 2000 patients, the overexpression of various SNHGs (especially SNHG1) is a poor prognostic factor in CRC (HR 1.64, 95% CI 1.40–1.86). The authors also presented all the signaling pathways interacting with this type of lncRNA. Many lncRNAs enhance cancer cell proliferation, acting directly or through different miRNAs [316]. For example, SNHG20 exerts this effect by directly affecting cyclin A1, and its expression is a poor prognostic factor in CRC (HR 2.97, 95% CI 1.51–5.82) [317]. Zhuang et al., in their meta-analysis, showed that the overexpression of lncRNA HNF1A antisense RNA 1 (HNF1A-AS1) could also be a recognized factor for poor prognosis (HR 3.10, 95% CI 1.58–6.11) [318]. A meta-analysis of numerous solid tumors (including CRC) revealed that increased expression of five prime to Xist (FTX) [314] and KCNQ1 opposite strand/antisense transcript 1 (KCNQ1OTI) correlated with shorter OS in CRC [315]. A previous study on metastasis-associated lung adenocarcinoma transcript 1 (MALAT-1/NEAT1) in six different tumors (including CRC) showed that the high expression of MALAT-1 correlated with lymph node and distant metastases (OR 3.52, 95% CI 1.06–11.71) [265]. In contrast, Xie et al. demonstrated a prognostic role for high levels of CRNDE in various cancer types including CRC (poor OS) (HR 2.11, 95% CI 1.63–2.75) [319].

Table 4 summarizes the examples of lncRNAs as prognostic markers in CRC, published in recent years.

In a previous review focused on the expression of some ncRNAs (miRNAs and lncRNAs), attention was drawn to the low reproducibility of the results and the poor power of statistical analyses for the reliability of the study. This may be due to both the small amount of material assessed or the over-sampling of ncRNAs, resulting in false positive or negative results [280]. The limitations of ncRNA detection in archival tissue material include high tumor heterogeneity, leading to an increasing preference to detect these molecules in serum/plasma or stool for prognostic purposes [276,303].

It should be noted that protein-coding mRNAs have a short half-life, and their expression changes enormously depending on the physiological/pathological state. Therefore, they are not ideal as prognostic indicators. The correlation between mRNA expression and protein translation is not always guaranteed, especially in heterogeneous tumors (including CRC), prompting the need for more sophisticated molecular techniques to assess the actual expression of biomarkers. In addition, studying complex interactions between different RNA types requires modern technology (e.g., high-throughput CLIP-Seq, degradome-Seq, and RNA–RNA interactome sequencing methods) [46,320].

#### 4.4.4. Prognostic Genetic and Epigenetic Biomarkers

The most relevant genetic and epigenetic alterations have been described as ‘potential prognostic markers’ [244] or ‘potential emerging biomarkers’ of clinical utility [321,322,323]. Initially, numerous panels of genes have been identified for metastatic CRC patients. For example, among the common core of five genes including *BRAF, EGFR*, *KRAS*, *NRAS*, and phosphatidylinositol-4,5-biphosphate 3 kinase catalytic subunit alpha (*PIK3CA*), two of them, *EGFR* and *PIK3CA,* have been named as ‘emerging biomarkers’ [321].

In an era of revolutionary advances in molecular biology techniques and bioinformatical methods, different strategies are being adopted to classify biomarkers in CRC. Considering the mechanisms of carcinogenesis, *KRAS, BRAF*, *APC,* and *TP53* genes have a permanent place among genomic biomarkers, whose role can be retrospectively traced to the Vogelstein model [7,45,245,324]. Mutations in protooncogenes (including *KRAS*) confer a strong growth signal to cancer cells and are closely associated with the development of CRC [5,55]. Notably, the *KRAS* mutation is currently the only marker with proven benefit for routine clinical use and selection for anti-EGFR mAbs therapy [324]. However, the presence of *KRAS* mutations does not always correlate with cell proliferation or the survival of patients with CRLMs [251]. In contrast, a meta-analysis by Sorich et al. showed that patients with metastatic CRC without *RAS* mutations (either *KRAS* exon 2 or new *RAS* mutation) treated with anti-EGFR mAbs had longer PFS and OS compared to patients with the presence of these mutations [325]. A recent study performed in a group of 73 CRC patients from South Korea reported no differences in DSF and OS treated with the FOLFOX regimen in groups divided according to the presence of *KRAS* mutations and the expression status of the excision repair cross-complementing 1 (ERCC1) protein. Interestingly, it was shown that the subgroup of patients with wild-type *KRAS* and increased IHC expression of the ERCC1 protein had lower OS compared to the subgroup with decreased ERCC1. No significant difference was found in the group of patients with mutated *KRAS*. In addition, the authors suggest that the presence of wild-type *KRAS* in combination with ERCC1 overexpression may be associated with oxaliplatin resistance. In other words, the *KRAS* status and ERCC1 expression in CRC patients treated with oxaliplatin-based CTx exhibit significant prognostic value [326].

An association has also been found between the loss of *TP53* (17q-*TP53*) and poorer survival rates, but *TP53* is not considered as a useful prognostic marker as the current data are insufficient to validate it [44]. Similarly, mutations in *TGF-β,* rare in CRC, cannot be indicated as significant prognostic factors in this cancer [5]. One meta-analysis showed a weak correlation between short OS and loss of 18q (HR 2.0, 95% CI 1.49–2.69), which encodes two crucial tumor suppressor genes (*SMAD2* and *SMAD4*) of the TGF-β family. Loss of function of these two genes leads, among others, to cell cycle deregulation [327]. In turn, the prognostic value of chromosomal instability in the form of CIN and MSI has been confirmed (also in meta-analyses) [245,327,328]. Erstad et al. listed genes including matrix metalloproteinases (*MMPs),* tumor inhibitor of metalloproteinase-1 (*TIMP-1),* manganese superoxide dismutase (*mnSOD)*, *TGF-β*, *Survivin*, and prolactin receptor (*PRLR*) among the prognostic factors of survival. From the classical proliferative markers, they mention the genes for TS and PCNA. The publication also provides an algorithm for the determination of prognostic biomarker profiles in CRC [244]. The following have also been cited as prognostic or predictive markers related to disease recurrence after surgery or resistance to treatment: ‘SC signature’ circulating tumor (ct)DNA and cell-free (cf)DNA, *RAS*, *PIK3CA* mutations, loss of *PTEN* (shorter PFS), low expression of *EGFR* (increase tumor regression), high density of TILs (better survival), loss of Bcl-2 expression (tumor recurrence), and somatic mutation of *BRAF* (mainly V600E) [323].

Epigenetic alterations in CRC mainly comprise abnormal methylated DNAs, abnormal histone modifications, and changes in the expression levels of abundant ncRNAs [329,330,331]. The prognostic significance of ncRNAs is described in Section 4.4.3. While the studied epigenetic aberrations in CRC include CIMP [50,323,332], opinions on the prognostic value of this marker differ and are debated by others [323,324].

Several DNA-methylation markers with prognostic value in CRC have also been demonstrated. These include the methylation of genes such as secreted frizzled-related protein (*SFRP*), *p16*, and long interspersed nucleotide element-1 (*LINE-1*). Methylation of *SFRP*, which acts as a tumor suppressor gene, is associated with increased CRC cell proliferation and tumor growth [333]. In turn, the methylation of *LINE-1* is associated with poor prognosis, shorter survival, and advanced stage [334,335]. A meta-analysis on *LINE-1* also suggests that its methylation is significantly related to the survival of CRC patients and may be a prognostic factor [336]. Another gene that undergoes DNA methylation in CRC is the DNA-binding protein Ikaros (*IKZF1*), which regulates the cell cycle. Methylation of the *IKZF1* promoter is associated with the loss of regulation of tumor cell proliferation and differentiation [337].

Recent studies also indicate high sensitivity and specificity in the detection of circulating DNA methylated in branched-chain aminotransferase 1 (*BCAT1*)/*IKZF1* in CRC compared to other cancers (breast, prostate) [338]. Other reviews additionally included methylated biomarkers of prognostic importance such as *p14*, Ras association domain-containing protein 1A (*RASSF1A*) and *APC* (poor prognosis), O-6-methylguanine-DNA methyltransferase (*MGMT*), DNA mismatch repair protein (*hMLH1*) (improved survival), homeodomain-only protein (HOPX-β) (worse prognosis in stage III CRC) and several EMC genes (worse survival), and IGF-2 hypomethylation (poor prognosis, short survival) [245,329]. Moreover, a recent study combined classical histopathology, the IHC method (p53 and Ki-67 expression), and MSP (aberrant methylation of *p16*, *E-cadherin*, *APC*, RUNX family transcription factor 3 (*RUNX3*), and *hMLH1*) with autofluorescence imaging (AFI) to assess the proliferative capacity of CRC. Abnormal expression of p53 and Ki-67 and the altered methylation of *p16* correlated with a lower AFI intensity [339].

It is important to note that the DNA methylation of genes in CRC also plays a role as predictive markers and/or can be a basis for the development of novel methylation-based therapies. Recent publications point to the important role of selected DNA methylation markers for the screening and early diagnosis of CRC [323,331,340]. One such plasma PCR-based test is the Epi proColon^®^, which is used to detect methylated *SEPT9* and has been approved by the U.S. Food and Drug Administration (FDA) for CRC screening in the U.S. The test, which is performed in conjunction with a stool test for methylated DNA from CRC cells, is used in patients who reject traditional screening methods [55].

Modern marker testing strategies in CRC potentially allow for the discovery of thousands of new genomic and transcriptomic factors. At least some of these are expected to become sensitive and specific proliferative markers with prognostic significance [55,323,324,340,341].

Considering the mechanisms of colorectal carcinogenesis associated with familial CRC, clinically useful markers such as dMMR, MSI, *KRAS*, *BRAF*, *APC, SMAD4,* and *BMPR1A* have already been indicated [47,323]. Markers with crucial roles in the pathogenesis of CRC also include key genes in the cell cycle process [324,328]. Moreover, a range of state-of-the-art molecular technologies used to detect a whole range of diagnostic markers in the human body (blood/plasma, tissue, stool) have also been described [340,342]. There are several recent summaries regarding the available technologies in for the search for the most sensitive, specific, low-cost, and reliable diagnostic, prognostic, and predictive markers in CRC [55,322,323,340,342].

Several publications have summarized the data on the clinical application of NGS technology in CRC [46,321,322,324,342,343]. Additionally, NGS allows for the identification of unknown interactions between genetic variation in CRCs and the relationship of CRCs to the structure of the gut microbiota composition [342].

Regarding the prognostic value in CRC, the activity of many protumoral genes (e.g., *CD74*, *CLCA1*, and *DPEP1*) has been described using this method. Additionally, using this technique revealed intra-tumor cell heterogeneity in ulcerative colitis (UC)-associated CRC [344]. Another study, based on several complementary techniques including scRNA-Seq, revealed that kinesin family member 21B (*KIF21B*) was highly expressed in CRC and was associated with poor survival. *KIF21B* expression was positively correlated with infiltrating CD4+ T cells and neutrophil levels, cell apoptosis, and metastasis. In vitro studies confirmed the role of *KIF21B* in enhancing proliferation, migration, and invasion [345]. In addition, one study based on scRNA-Seq, RNA-Seq, and microarray cohorts established a prognostic model based on the composition of prognosis-related cell subsets in TME including nine specific immune cell lineages [346].

Other publications have described the advantages and technical challenges of using liquid biopsies in the form of circulating tumor cells (CTCs) [347] and ctDNA as a promising alternative to molecular tissue analysis [323,348,349]. ctDNA detection in blood can be used to predict CRC recurrence after surgical resection [323]. In turn, a meta-analysis (2016) showed strong associations between cfDNA, RFS (HR 2.78, 95% CI 2.08–3.72), and OS (HR 3.03, 95% CI 2.51–3.66) in CRC patients. Thus, the appearance of cfDNA in the blood can predict shorter OS and worse RFS regardless of the tumor stage, study size, tumor markers, detection methods, and marker origin [349]. Nonetheless, targeted NGS analysis of cfDNA from TruSight Tumor 170 (TST170) may be useful for the non-invasive detection of gene variants in metastatic CRC patients. TST170 is an NGS panel that covers 170 cancer-related genes including *KRAS* [350]. High compatibility was also detected between cfDNA and tumor DNA in metastatic CRCs using a 10-gene NGS panel. *TP53* was the most frequently mutated gene (63.2%), followed by *APC* (49.5%), *KRAS* (35.8%), and FAT tumor suppressor homolog 4 (*FAT4*) (15.8%). The concordance of mutation patterns in these 10 genes was as high as 91% between the cfDNA and tumor samples. These results also confirmed the high sensitivity (over 88%) and specificity (100%) of the *KRAS* status in cfDNA for predicting mutations of this gene in tumor tissue. Significant prognostic correlations (peritoneal, lung metastasis) between *TP53*, *KRAS,* and *APC* mutations in the tumor were also demonstrated [351].

The use of ctDNA-based genotyping of *KRAS*, *NRAS,* and *BRAF* indicates the utility of predicting patient survival depending on the mutations of these genes. The highest mutation frequency is attributed to *KRAS* (34%). The median OS of patients with *RAS/BRAF* mutations detected in plasma was 26.6 months, and patients with wild-type *RAS/BRAF* did not reach median survival during follow-up. The median RFS for *RAS/BRAF* wild-type and *RAS/BRAF* mutation patients was 12 and 4 months, respectively [352]. Attempts have been made to determine the prognostic role of markers detected by CTC-based techniques in CRC. However, the results are still not convincing enough for recommendation in clinical practice [340,347].

A recent review paper summarized the use of various molecular techniques (e.g., RT-PCR, PCR, and single nucleotide polymorphism (SNP) genotyping assay, NGS, NanoString analysis, Sanger sequencing, MassArray sequencing, quantitative MSP) to investigate changes at the DNA and RNA level that may predict CRC metastasis to the peritoneum. Only *BRAF* mutations were associated with peritoneal metastases in 10/17 studies [353].

The development of modern, especially non-invasive molecular technologies in CRC should improve the specificity of tests (above 90%) primarily for disease screening and therapeutic decisions [55,341,354].

Nowadays, numerous molecular techniques can be chosen, and the decision to use specific markers should balance advantages and limitations that may affect the final results. In the case of NGS-based DNA nucleotide variation testing, the difficulty lies in wide variety of NGS platforms and gene panels and the multi-step nature of the study. In terms of sensitivity, ctDNA NGS techniques cannot compete with digital PCR, prompting the need for PCR result validation. The sensitivity issue is important, particularly in liquid biopsy, as random results (false positive mutations) can be obtained from hematopoietic clones rather than from the tumor itself. The significant number of gene variants of unknown roles obtained in the study is also not accepted by experts due to the lack of clinical utility [321,348]. Another author described the limitations of non-standardized methods, the small cohorts of patients analyzed, and the lack of demonstration of a clear clinical benefit of liquid biopsy studies in CRC [347]. The scientific literature also contains proposals for a systematic review and meta-analysis protocol in detecting *KRAS* mutations in CRC using liquid biopsy samples, with paired tissue samples serving as the control [355].

### 4.5. Positron Emission Tomography (PET) to Assess Tumor Growth Rate

PET is the most specific and sensitive method of in vivo molecular interaction and pathway imaging, finding an increasing number of applications in oncology [356]. This non-invasive technique for the functional imaging and assessment of CRC growth rate is based on the use of labelled 18-fluoro-3-deoxy-3-fluorothymidine (FLT). The method can reveal the spatial organization of proliferating cells in the tumor and allows for multiple simultaneous in vivo measurements. However, there are some correlations between FLT uptake and tumor proliferative activity [111,112]. FLT was reported to have high sensitivity in detecting extrahepatic disease but poor sensitivity in imaging CRC liver metastases [112]. A better and currently the most commonly used tracer in CRC is 18F-labelled 2-fluoro-2-deoxy-D-glucose (18F-FDG), with its usefulness resulting from increased glucose consumption by malignant cells. Therefore, this tracer’s uptake is closely linked to cancer cell proliferation, which depends mainly on glycolysis for energy. Many signal transduction pathways in the malignant transformation of cancer cells are regulated by glycolytic metabolism [357]. Therefore, the combination of PET and 18F-FDG has become an established tool for diagnostic tumor imaging and complete preoperative staging in CRC [112,113,358]. PET–18F-FDG results may have implications for the therapeutic management of patients with CRC [358,359] including metastatic CRC [113]. One review recognized that PET in CRC also allows for the metabolic characterization of lesions suspected of recurrence or the identification of latent metastatic disease [358]. Comparative studies indicate lower FLT versus FDG uptake in patients with CRC. However, no correlation was shown between the two radiotracers used and the proliferative activity assessed by the Ki-67 index [360]. A later meta-analysis only confirmed a moderate correlation between 18F-FDG uptake and Ki-67 expression in CRC [361]. A recent study by Watanabe et al. indicated that tumor proliferation in CRLM is reflected by the standardized uptake value (SUV) from FDG-PET. In addition, the authors showed a high correlation between SUV and Ki-67 expression. SUV was also shown to include factors of glucose metabolism (expression of hypoxia-inducible factor 1 alpha (HIF-1α), pyruvate kinase M2 (PKM2), and glucose transporter 1 (GLUT1)). Thus, this test can be a valuable method to assess the proliferative and metabolic viability of the tumor in advanced CRLM [113].

The remaining limitations of PET comprise its high cost and the lack of necessary equipment in cancer centers, limiting the potential for multidisciplinary PET studies. The most significant limitation for the patient is the need for the administration of radioactive tracers, resulting in potential radiation exposure [356].

Figure 1 summarizes the major categories of prognostic proliferative markers in CRC and the most important signaling pathways that are genetically altered in CRC progression.

## 5. Final Remarks and Future Perspectives

The development of IHC and modern molecular biology techniques (qRT-PCR, ISH, RNA/DNA sequencing, NGS, DNA methylation detection methods) has made it possible to determine the prognostic efficacy of many classic IHC markers for the estimation of patient survival, disease-free time, or disease recurrence. The prognostic role of aneuploidy, overexpression of markers such as TS, cyclin B1 (better 5-year survival), cyclin D1 (poor OS and DFS), PCNA (poor OS and CSS), and Ki-67 (poor OS) could be confirmed. Ki-67 antigen was also among 26 independent biomarkers of OS in resected CRLMs. However, studies indicating the overexpression of Ki-67 or other proliferation markers as good predictors of survival remain controversial. It has been suggested that an association between a high Ki-67 index and improved survival is only present in MSI-H status tumors. In turn, RT-PCR studies showed a high positive correlation of Ki-67 mRNA with pKi-67 and confirmed the role of Ki-67 mRNA as a predictor of poor OS. Studies indicate that tumors with high pKi-67 and low mRNA levels are likely to proliferate more slowly and have a better prognosis.

In CRC therapy (especially RT), slowly proliferating cells should also be considered, in addition to rapidly proliferating cells. Such cells provide a reservoir from which cells can be recruited to short-cycle, resulting in accelerated cell repopulation in response to damaging factors (irradiation, hypoxia). However, further research is required to clarify to what extent the pool of slowly proliferating cells includes CSCs that may reside in the G0 phase. So far, an optimal panel of IHC assays with markers of cellular proliferation in CRCs has not been established to predict survival or the effect of adjuvant treatment. While Ki-67 shows some promise as one of the components of the Oncotype Dx Colon Cancer Assay for predicting the risk of recurrence in stages II and III colon cancer, recent studies do not recommend this assay for use in patients with stage II CRC.

Modern molecular biology techniques have confirmed or discovered the role of several genetic and epigenetic markers, mainly as diagnostic and predictive markers in CRC. New technology also allows for the identification of a broad range of candidate prognostic markers. Classic genetic markers of prognostic significance include mutated genes (e.g., *APC*, *KRAS/BRAF*, *TGF-β,* and *TP53)*, chromosomal markers CIN and MSI, epigenetic markers such as CIMP, and many other candidates including *SERP*, *p14*, *p16*, *LINE-1*, and *RASSF1A*. Further research is required to determine the prognostic role of *KRAS* mutation status in different CRC patient populations worldwide. Similarly, continued research is necessary to determine the contribution of *KRAS* mutations to the mechanisms of drug resistance to oxaliplatin.

The number of long non-coding RNAs (e.g., SNHG1, SNHG6, MALAT-1, CRNDE) and microRNAs (e.g., miR-20a, miR-21, miR-143, miR-145, miR-181a/b) related to proliferation in CRC as confirmed prognostic markers is also increasing. Despite the rather obvious limitations of IHC and new molecular techniques, the standardization of methods for the quantitative assessment of the expression of proliferation markers, or the understanding of endogenous and exogenous (environmental) mechanisms of accelerated cellular proliferation, requires further development. For a more accurate survival prognosis or prediction of therapeutic effects in CRC, it would be ideal to use complementary methods to study cell cycle disruption, apoptosis, and genomic alterations. The expanding development of research techniques is undisputedly contributing to the systematization of knowledge regarding cancer biology. Moreover, the detection of numerous ncRNAs, given their role in cell cycle regulation in CRC, cannot be underestimated. However, the recommendation of a specific ncRNA or a panel of such molecules as clinically useful prognostic markers is still a matter of the future. The previously signaled need to validate large-scale research and conduct multicenter studies on different populations will help to create a base of more reproducible results and identify their potential application in CRC patients.

## 6. Conclusions

As the reviewed literature reports, the prognostic utility of aneuploidy testing and of some immunocytochemical markers of cellular proliferation in CRC (TS, cyclin B1 and D1, PCNA, and Ki-67) needs to be supplemented by modern molecular biology techniques. The limits of conventional techniques to assess cellular proliferation, the high heterogeneity of tumor tissues, etc., justify the search for a panel of optimally sensitive, specific, and non-invasive CRC biomarkers. A specific expression pattern of ncRNAs (miRNAs and lncRNAs) may prove helpful in effectively identifying patients with a poor prognosis. It is particularly important to confirm known gene mutations/epigenetic alterations and to identify new mutation ‘patterns’ in different CRC patient populations to determine the prognosis for survival and/or the effects of cytotoxic and biologic regimens. For clinical and personalized medicine purposes, it seems important to construct a commercial test, based on a broad, prospective study, with the independent validation of biomarkers with prognostic/predictive value.

## Figures and Tables

**Figure 1 cancers-15-04570-f001:**
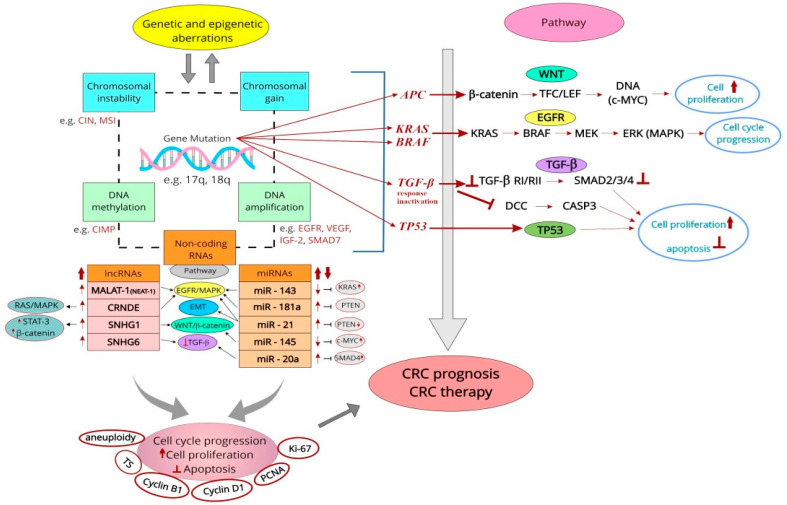
Different categories of proliferative markers with potential prognostic and predictive significance in CRC in association with key signaling pathways that are dysregulated in the ‘adenoma–carcinoma sequence.’ Dysfunction of the WNT/β-catenin, EGFR/MAPK, TGF-β, and TP53 signaling pathways leads to cell cycle progression, increased proliferation, and the inhibition of cell apoptosis. Classic genetic markers of prognostic significance include the mutated genes (e.g., *APC*, *KRAS/BRAF*, *TGF-β,* and *TP53*). Markers at the chromosome level are CIN and MSI. Epigenetic markers are CIMP and many other candidates including *SERP*, *p14*, *p16*, *LINE-1*, and *RASSF1A* (not shown). Selected genes also undergo amplifications (e.g., *EGFR*, *VEGF*, *SMAD7*, *IGF-2*), enhancing cellular proliferation. Prognostic markers also include ncRNAs. Several representatives were selected based on their proven role in cell cycle progression and enhancement of proliferation, with prognostic value demonstrated in meta-analyses. A prognostic role for aneuploidy and altered expression of conventional IHC proliferating markers in CRC (i.e., TS, cyclin B1, cyclin D1, PCNA, and Ki-67) was shown. Legend: ⇓—regulation; ↑/↓—increase/decrease; Ʇ—inhibition; APC—adenomatous polyposis coli; BRAF—protooncogene B-Raf; CASP3—caspase 3; CIMP—CpG island methylator phenotype; CIN—chromosomal instability; c-Myc—protooncogene from Myc family; CRC—colorectal cancer; DCC—deleted in colorectal cancer; EGFR—epidermal growth factor receptor; EMT—epithelial–mesenchymal transition; ERK/MAPK—extracellular signal-regulated kinase or classical MAP kinase; IGF-2—insulin-like growth factor 2; IHC—immunohistochemical; KRAS—Kirsten rat sarcoma virus; LINE-1—long interspersed nucleotide element-1; MEK/MAP2K—mitogen-activated kinase; MSI—microsatellite instability; ncRNAs—non-coding RNAs; PCNA—proliferating cell nuclear antigen; PTEN—phosphatase and tensin homolog deleted on chromosome ten; RAS–rat sarcoma virus, three Ras genes in humans: HRAS, KRAS, and NRAS; RASSF1A–Ras association domain-containing protein 1A; STAT3—signal transducer and activator of transcription 3; TGF-β (RI/RII)—tumor growth factor β (receptor I/II); TS—thymidylate synthase; SERP—secreted frizzled-related protein; SMAD2/3/4/7—mothers against DPP homolog 2/3/4/7; VEGF—vascular endothelial growth factor; WNT—gene wingless *+* integrated or int-1.

**Table 1 cancers-15-04570-t001:** Cyclins and their potential prognostic value in colorectal cancer (CRC).

Type of Cyclin	Material (No. of Cases) and Method	Findings	Year of Publication	Ref. No.
A (A2)	CRC (73); IHC, SI	Mean: 12.26 ± 5.8; SI was correlated with tumor differentiation; ↑expression correlated with ↓OS; ↑expression is an independent negative prognostic factor (HR 7.82, 95% CI, 0.02–60.12) (UA) and (HR 13.89; 95% CI 1.01–190.58) (MA)	1999	[159]
	CRC (60); IHC, SI	↑Expression associated with ↓OS; independent prognostic factor	2004	[160]
	CRC (167); IHC	(+) Expression (61.1%); (+) expression correlated with ↓survival; (+) expression, LN meta, and Dukes’ stage were independently associated with unfavorable prognosis	2004	[162]
	CRC (219); IHC; qRT-PCR	(+) Expression (83%), extra gene copies (6.2%); correlation with stage and differentiation; ↑expression independently associated with improved survival (UA), (HR 0.57, 95% CI 0.33–0.98) (MA)	2005	[155]
	CRC (790); IHC	Expression above the median predicted an improved patient prognosis (HR 0.71, 95% CI 0.53–0.95); cell proliferation and (+) expression were prognostic indicators of patient outcome	2011	[163]
B (B1)	C (22), ADs (62); CAs in ADs (17), pCRC (194), LN meta (21); IHC	↑B1 expression from C through ADs to pCRC; ↑expression with increasing degree of dysplasia in ADs, from peripheral ADs to central CAs, and from primary to metastatic foci; ↓in pCRC with large size, mucinous type, deep invasion, or short PPS time	2003	[164]
	CRC (342); IHC	↑Expression (78.7%); no association with histopathologic features; no impact on OS and DFS (UA)	2004	[165]
	CRC (219); IHC, qRT-PCR	(+) Expression (83%), extra gene copies (9%); no prognostic value	2005	[155]
	CRC (150); WB; qRT-PCR; IHC	↑mRNA expression (92.7%); ↑expression negatively related to LN and distant meta, and TNM; ↓expression associated with poor OS	2015	[166]
C	CRC (219); IHC; qRT-PCR	↑Expression (88%), extra gene copies (26.9%); ↑expression correlated with *CCNC* amplification; protein expression tends to associate with DSS; *CCNC* amplification related to an unfavorable prognosis, (HR 1.72, 95% CI 1.00–2.94) (MA)	2005	[155]
D (D1, D3)	CRC (123); IHC	↑D1 expression correlated with poor OS and DFS; an independent predictor of disease recurrence	1998	[172]
	CRC (90); IHC	Nuclear/cytoplasmic expression of cyclin D1; no prognostic value	1998	[177]
	CRC (73); IHC, SI	Mean: 6.9 ± 6.3; SI correlated with tumor differentiation; ↑in LN meta vs. those without; ↑in advanced than in early CAs; ↑expression tends to associate with poor prognosis	1999	[159]
	CRC (126); IHC	(+) Expression (58.7%); cytoplasmic (HR 0.56, 95% CI 0.31–1.0) or nuclear level (HR 0.24, 95% CI 0.07–0.81) related to ↑survival	2001	[183]
	RC (160); IHC, (+) at the 10% level	(+) D1 expression (48%); no prognostic role of this marker	2002	[178]
	CRC (60); IHC, SI	↑SI within deeply invasive tumors and LN meta; ↑expression and D1 amplification associated with ↓OS; independent prognostic factor	2004	[160]
	CRC (219); IHC; qRT-PCR	(+) D1 expression (11%) and extra gene copies (55%), cyclin D3 (36%) and extra gene copies (20.5%); no prognostic role of these markers	2005	[155]
	CC and RC (363), Dukes’ A–D, TMA; IHC	(+) Nuclear staining of cyclin D1 reflected better survival	2005	[182]
	CRC (97); IHC, (+) at >5% cells	↑Expression (5.9%); ↑levels in mucous differentiation; ↑expression correlated with stage, LN meta; no prognostic value	2008	[179]
	CC (602), stage I–IV; IHC	↑Expression (55%) was related to low cancer-specific mortality (HR 0.57, 95% CI 0.39–0.84) (MA), and for low overall mortality (HR 0.74, 95% CI 0.57–0.98); ↑expression related to ↑survival	2009	[181]
	CRC (84), TMA; cyclin D1, D2, D3; IHC	D2 expression at the margin associated with vascular invasion, LN meta, and CRLM; ↑D2 and D3 associated with vascular invasion, CRLM, and ↓DSS (cyclin D2)	2010	[173]
	CRC (169); IHC	(+) D1 expression related to shorter survival	2011	[174]
	CRC (117), TMA; IHC	↓Nuclear expression associated with negative lymphovascular invasion; no prognostic value of cyclin D1	2015	[180]
	CRC with meta (1205); IHC	↑Expression (46.7%); ↑D1, EGFR expression, late stage after S indicated ↓RFT (UA); no independent factor of prognosis (MA)	2019	[175]
	CC (102); IHC	(+) Expression of cyclin D1 correlated with a worse 5-yrs survival rate in pts with advanced stage (III, IV)	2019	[169]
	CRC (101), Dukes’ B and C stages; IHC	↑Expression more often in DFS ≤24 group vs. ≥48 group and had 5.2 higher odds of having DFS ˂24 mo; ↑expression correlated with early recurrence in high-risk Duke’s B and C stage	2021	[176]
E	CRC (219); IHC, qRT-PCR	(+) Expression (25%) and extra gene copies (19.1%); no prognostic value	2005	[155]
	CRC (97); IHC, (+) at >5% cells	↑Expression (30%); (+) correlation with p21waf1/cip1, PCNA-LI and Ki-67; no prognostic value	2008	[179]
	CRC (200), benign alterations (200); IHC; RT-PCR	↑Expression with TNM and decreasing tumor differentiation; (+) expression correlated with shorter PFS and median survival	2016	[185]
	CRC (31), TMA; IHC	(+) Expression (34.78%); no prognostic role of this marker	2016	[186]
	CC (102); IHC	(+) Correlation with cyclin D1; no prognostic role	2019	[169]

Legend: ↑/↓—increase (overexpression)/decrease; </>—lower/higher; (+)/(−)—positive/negative; ADs—adenomas; C—control, normal mucosa; Cas—carcinomas; CC—colon cancer; *CCNC*—cyclin-C encoded gene; CI—confidence interval; CRLM—CRC liver metastasis; DFS—disease-free survival; DSS—disease-specific survival; (p)CRC—(primary) colorectal cancer; EGFR—epidermal growth factor receptor; HR—hazard ratio; IHC—immunohistochemistry; LI—labeling index; LN(s)—lymph node(s); MA—multivariate analysis; meta—metastasis; mo—months; no.—number; OS—overall survival; PCNA—proliferating cell nuclear antigen; PFS—progression-free survival; PPS—postoperative patient survival; pts—patients; RFT—relapse-free time; qRT-PCR—quantitative real-time polymerase chain reaction; RT-PCR—reverse transcriptase-polymerase chain reaction; S—surgery; SI—staining index; UA—univariate analysis; WB—Western blot analysis; yrs—years.

**Table 2 cancers-15-04570-t002:** Prognostic value of proliferating cell nuclear antigen (PCNA) in colorectal cancer (CRC).

S No.	Material (No. of Cases) and Methods	Findings	Prognostic Role	Year of Publication	Ref. No.
1.	CRC (40); IHC, PI	↑PI in both cancer and epithelial cells of adjacent C crypts in those who died vs. survivors	PI is an independent predictor of recurrence and poor survival in both groups	1993	[189]
2.	CRC (82) and LN meta (18); IHC, q estimation	Similar to the median and range of the % of (+) cells in primary tumors and LN meta	An inverse relationship between the % of (+) cells and survival times	1993	[192]
3.	CRC (60) and ADs (35); IHC; FCM for DNA content	Mean: 38% (ADs); mean: 50.4% (CRC); aneuploid ACs had a tendency to poorer prognosis, especially in Dukes’ C female pts	Can be an indirect indicator of cells in the S phase; is not an independent prognostic factor	1994	[123]
4.	CRC (125); IHC, LI, image analysis	LI without significant correlation with clinical characteristics (stage, grade, age, sex, fixity)	No prognostic role for survival	1994	[124]
5.	CRC (49); IHC, LI	↑LI of tumors with venous invasion (mean: 51.7%); with LN meta (mean: 50.5%); with meta to the liver (mean: 55.2%); ↑LI associated with less differentiated tumors	Evaluation of LI at the invasive tumor margin may help identify CRC with ↑malignant potential	1994	[193]
6.	CRC biopsies (50); FCM, LI	LI from 38.7% to 53.0%; in diploid tumors (27), the median LI in G0/G1: 71.5%, in S: 10.5%, in G2/M: 17.4%	Is expressed throughout the cell cycle; prognostic role—probable	1995	[108]
7.	CC (50) and 40 RC; IHC (79), LI	LI improved the prediction of survival when used with histopathological classification (Dukes’ or Jass’) (MA)	Little prognostic power of LI (UA); not predictive for RC; ↓LI related to the worst prognosis	1995	[191]
8.	CRC (57); IHC, LI	↑Deep invasion, CRLM, and ↑stages with ↑LI (>49.4%) vs. ↓LI; ↑survival curves for pts with (−) CEA and ↓LI vs. pts with (+) CEA and ↑LI; ↑survival curves for pts with (+) CEA and ↓LI vs. pts with (+) CEA and ↑LI	Serum CEA and PCNA LI for cancer pts are useful in the evaluation of tumor progression and prognosis	1996	[194]
9.	CRC (86); IHC, LI	↑LI with stage, histologic differentiation, lymphatic and vascular invasion, LN meta, and CRLM; ↑LI in tumors with DNA aneuploidy	↑Recurrence rate with LI > than the mean LI; ↓4-yr survival rates for overall and curative pts with LI > than the mean LI	1996	[190]
10.	CRC (59); IHC, LI	Lesions combining HLA-DR expression and a relatively ↓LI had the best prognosis	HLA-DR expression with ↓LI is an important outcome predictor	1998	[196]
11.	CRC (47); IHC, >60% nuclei (+)	(+) Correlation with Bcl-2, LN meta, and tumor location	May be an indicator of the development of LN meta	2009	[197]

Legend: ↑/↓—increase/(overexpression)/decrease; (−)/(+)–negative/positive; </>—lower/higher; AC(s)—adenocarcinoma(s); AD(s)—adenoma(s); AS1—antisense to PCNA; C—control, normal mucosa; CRLM—CRC liver metastasis; FCM—flow cytometry; HLA-DR—human leukocyte antigen–DR isotype, major histocompatibility complex, class II, DR alpha; HP(s)—hyperplastic polyp(s); LN(s)—lymph node(s); meta—metastases; No.—number; PI—proliferating index; pTNM—pathological tumor/node/metastasis; pts—patients; q—quantitative; ROC—receiver operating characteristic curve; RR—risk ratio; S No.—study number.

**Table 3 cancers-15-04570-t003:** Immunohistochemical (IHC) studies on the prognostic relevance of Ki-67 in colorectal cancer (CRC) and CRC with liver metastases (CRLM) (S No. 9, 13, 26, 29, 32).

S No.	Material (No. of Cases) and Methods	Findings	Prognostic Role for Survival	Year of Publication/Country	Ref. No.
1.	pAC (139); IHC, mAb Ki-67, LI; S	↑In mucinous vs. non-mucinous CRC; inverse correlation with grading in non-mucinous AC	No	1990; Italy	[231]
2.	pCRC (125); IHC, LI; S	No correlations with clinicopathological data	No	1994; UK	[124]
3.	CRC (106); IHC, MIB-1, 3 methods of estimation; S	No correlation with clinical outcome	No	1996; Austria	[233]
4.	CRC (70); stages II and III; IHC, MIB-1, LI; S	Relation to disease recurrence, retained in stage II; LI > 45% associated with ↑risk for disease recurrence vs. LI ≤ 45% (MA)	Yes	1997; USA	[125]
5.	CRC (255); Dukes’ A–D; IHC, MIB-1, weak (<50%), strong (>50%); S	Level > 50% (62%); <50% (38%); no correlations with clinicopathological variables	No	1997; Sweden	[234]
6.	CRC (56); Dukes’ B; survival analysis (47); IHC, anti-Ki-67, morphometry; S	Mean value in luminal border (27.4%), invasive margin (36.8%); ↓LI at the invasive margin correlated with poorer survival (RR 12.1, 95% CI 1.1–1.33) (UA and MA)	Yes	1999; Sweden	[247]
7.	CRC (52); AD (56); IHC, MIB-1, LI; S	↓LI in AD (30.05%) vs. CRC (38.12%); ↑correlated with poor differentiation and Duke’s stage	Nd	2000; USA	[218]
8.	CRC (30); IHC, LI; S	No correlation with tumor stage and grade	Nd	2001; France	[127]
9.	CRLM (41); IHC, MIB-1; LI at the hot spot; S	Mean value (38%); LI ≥50% related to shorter survival vs. low scores; ↑score an independent adverse prognostic factor (RR 3.04) (MA)	Yes	2001; Germany	[251]
10.	CRC (25); pTNM; stages I–IV; IHC, MIB-1, LI, morphometry; RT-PCR; S	Median protein LI (61%), median mRNA LI (0.88 amol); better OS for the group with ↓LI and ↓mRNA level vs. median	Yes	2001; Germany	[259]
11.	CRC (100); MSI-H (31), MSI-L (29), MSS (40); IHC, PI; S	↑PI (90.1%) in MSI-H vs. MSI-L (69.5%) and vs. MSS (69.5%); ↑PI showed a trend toward predicting ↑survival only within MSI-H cancers	Probably yes	2001; Australia	[260]
12.	CC (465); Dukes’ B2 and C; S alone (151) or S + FU-based CTx (314); IHC, LI	No significant association with clinical outcome	No	2002; USA	[142]
13.	pCRC (74); CRLM (37); IHC, MIB-1, LI; S	LI ≥30% more frequently in lymphatic and venous invasion, LN meta, and CRLM; ↑in primary tumors vs. CRLM (24.3 ± 17.9 vs. 5.0 ± 4.2); LI ≥30% in pCRC correlated with ↑frequency of metachronous CRLM	Nd	2002; Japan	[221]
14.	CC (706); stages II and III; S alone (275) or S + FU-leucovorin CTx (431); IHC, LI	Tumors with ↑number of (+) cells had improved outcomes vs. tumors with few (+) cells; association with RFS (RR 0.76) and with OS (RR 0.62)	Yes	2003; USA	[147]
15.	CRC (47); IHC, MIB-1, LI; ISH for mRNA with DIG-labelled cRNA probe, LI; S	Median protein LI (59%), mean mRNA LI (42%); ↑protein but ↓mRNA are likely to proliferate more slowly, which possibly explains the pts’ improved outcome	No	2003; Germany	[236]
16.	CRC (81); IHC, anti-Ki-67, IRS; S	↑Expression in the low differentiated tumors; inverse correlation to survival	Yes	2003; PL	[222]
17.	pCRC (311 including 82 with distant meta); AD and CA in situ (22); IHC, MIB-1; S	↑Rate in AD with severe atypia and CA in situ; ↓rate in poorly differentiated and mucinous AC vs. well- and moderately differentiated tumors	Nd	2003; Japan	[219]
18.	CC (144), RC (90); IHC; MIB-1, semiq estimation; S	↑In LN meta of short-term (505 d) vs. long-term survivors (4150.5 d)	No, but an indicator of survival in Dukes’ C	2004; Japan	[240]
19.	RC and rectosigmoid AC (146); IHC, MIB-1, high (>40%) and low (≤40%), hot spot areas (>50%); S	Better OS for ↑values vs. those with ↓values; the presence of hot spot areas associated with better survival (MA); hot spot areas one of the prognostic factor	Yes	2005; Finland	[248]
20.	CRC (106); IHC, MIB-1, PI; S	Mean PI (38.0%); (+) correlation with advanced T status, LN and distant meta, and ↑pTNM stage; an independent prognostic factor for long-term survival; pts with high PI were at greater risk for death (HR 2.1, 95% CI 1.1–4.1) (MA)	Yes	2005; Japan	[223]
21.	CC (53), RC (33); stages I–IV; IHC, MIB-1, group A (<40%) and B (≥40%); S	Mean LI (0.44 ± 0.16); no correlation with sex, age, and clinical stage; ↑level correlated with ↓survival; an independent predictor of survival (MA)	Yes	2005; Brazil	[96]
22.	CRC (pCC + pRC) (363), Dukes’ A–D; IHC, LI; S	In RC, pts with a LI ≥5% had a better prognosis than those with a lower index	Yes	2005; Finland	[182]
23.	CRC (40); IHC, NCL-Ki67p, PI; S	Mean PI (52.39%); pts who developed either local recurrence or meta had a significantly raised PI; PI ≤52.7% with a trend to improved survival	No for OS (MA)	2006; UK	[261]
24.	CRC (38): mucinous (14), non-mucinous (24); stage B1, B2, C1, C; IHC, anti-Ki-67, hot spot, NIH’s Image I; S	Median (35%); (+) correlation with age, LN meta, and with Dukes’ MAC staging (25% in B1, 60% in C2); ↑with grade	Nd	2007; Romania	[224]
25.	CRC (47): mucinous (5), non-mucinous (42); pT3, G2; IHC, MIB-1, negative <50%; positive > 50%; S	(+) Correlation with LN meta	Nd	2009; PL	[197]
26.	CC (40), rectosigmoid or rectal AC (33); CRLM (27); IHC, MIB-1; qRT-PCR; S	pCRC (81.8%) vs. CRLM (36.2%); ↓of the GPS in CRLM and confirmed their ↓proliferative levels by qRT PCR	Nd	2009; New Zealand	[115]
27.	CRC (152), stages I–IV; IHC, rabbit anti-Ki-67, semiq estimation; S	(+) Correlation with the UICC stage and differentiation; (+) pts had the ↓cumulative survival vs. pts with no expression (MA)	Yes	2010; China	[225]
28.	CRC (356); IHC; S	No association with clinicopathological variables	Nd	2010; Korea	[235]
29.	CRLM (188/124 for Ki-67); IHC; S	↑Expression (62%); ↑expression as an independent predictor of poor survival after colon resection (HR 2.6, 95% CI 1.4–4.8)	Yes	2010; USA	[252]
30.	CRC (201); stages I–IV; IHC, anti-Ki-67, semiq estimation, (+) (score ≥5); S	(+) Expression (59.7%); (+) correlation with tumor size, grade, invasive depth, LN meta, distant meta, TNM; independent prognostic factor of favorable OS (HR 0.34, 95% CI 0.16–0.72) (MA)	Yes	2011; China	[249]
31.	CRC (31), men with Dukes’ B AC; IHC, MIB-1, semiq estimation; S	Median (46.9 ± 19.2%); inverse relationship with OS (r = −0.67)	Yes	2012; Italy	[241]
32.	TMA with CRLM (98); IHC, MIB-1; cut-off value for (+) phenotypes (>50%); S	More (+) pts among the long-term survivors; pts with ↑ expression lived longer (HR 0.82, 95% CI 0.68–0.98) (MA); positive predictor for AS, but not for DFS	Yes	2014; Slovenia	[238]
33.	RC (111); IHC, MIB-1, LI; SCRT + S	↑Expression correlated with pTR; in females (+) correlation with pTNM in a long break after SCRT	Nd	2014; PL	[88]
34.	TMA CRC (672), including CRC with LN meta (210); IHC, anti-Ki67, LI; S	Median in pCRC (68.2%), in LN meta (55%); ↑in pCRC vs. CRC LN meta; (+) correlation with tumor penetration and differentiation	No	2015; Portugal	[226]
35.	CRC (110) including Dukes’ C; IHC, MIB-1, LI; semiq estimation; S	↑Expression in LN meta vs. pCRC	No	2015; Turkey	[232]
36.	CRC (74) including mucinous AC (5); IHC, LI; S	LI of well (14%), moderate (31%), and poorly differentiated AC (43%); (+) correlation with stage and grade	Nd	2015; India	[227]
37.	CRC (2233), I–IV stage; IHC, MIB-1, low (<50%), high (≥50%); S	Pts in stage III with ↑level had ↑3-yr DFS and OS vs. ↓level pts; improved 3-yr PFS for stage IV pts in the ↑vs. ↓level group	Yes	2016; Germany/pts of Chinese origin	[250]
38.	TMA CRC (1800); IHC, anti-Ki-67, low (0–10%), moderate (>10–25%), high (>25%); S	↑Expression associated with low stage and LN status; an independent prognostic factor of favorable survival	Yes	2016; Germany	[237]
39.	TMA CRC (254), stage II and III; IHC, anti Ki-67, low (<20%) and high (≥20%); S	↑LI associated with ↑TNM stage; ↓LI related to RFS (UA); ↑LI (HR 2.62, 95% CI 1.12–6.14; an independent predictor of unfavorable prognosis (MA)	Yes	2018; China	[228]
40.	RC (46), stage II and III; IHC, MIB-1, Image System (Nikon), LI, cut-off value (30%); CRT + S	No difference between ↓ and ↑expression groups in clinicopathological factors; ↑LI correlated with lower 5-yr DFS vs. group with ↓LI (53% and 88%), as was the 5-yr OS (68% and 100%)	Yes	2018; Japan	[254]
41.	CRC (1090), stage 0-IV; IHC; anti-Ki-67, semiq estimation; cut-off value of 25%; S	(+) Correlation with invasive depth, differentiation, and size, AJCC-8, (+) no. of LN and CTx status; ↑level related to poor prognosis and independently predicts prognosis in the AJCC-8; no differences for DFS and OS in stage IV	Yes	2020; China	[229]
42.	CRC (38), non-neoplastic polyps (2) and AD (20); IHC, anti-Ki-67, LI; S	CRC: ↑LI in higher grade and stage; AD: ↑intensity and high score similar to CRC; non-neoplastic polyps: ↑LI and medium intensity; ↑LI from non-neoplastic to neoplastic cases	Nd	2021; India	[217]
43.	CRC (210), stages I–III; IHC, polyclonal Ab, LI, cut-off value 60%; S	LI ≥60% indicated a high-risk ratio for both distant meta (HR 2.56, 95% CI 1.08–6.06) and death (HR 2.64, 95% CI 1.07–6.54)	Yes	2022; China	[230]
44.	RC (154), RC I–II after RT + S (2–3 d after) (64), RC I–III after S (90); IHC, image analysis application package	↑Level with a survival rate of less than 3 yrs in both pts after RT and S	Yes	2022; Switzerland, Germany, UK	[255]

Legend: ↑/↓—increase (overexpression)/decrease; >/<—higher/lower; (+)—positive; AC—adenocarcinoma; AD(s)—adenoma(s); AJCC—American Joint Committee on Cancer 8th edition; AS—actual survival; C—control; CA—carcinoma; CC—colon cancer; CI—confidence interval; CRT—chemoradiotherapy; CTx—chemotherapy; d—days; DFS—disease-free survival; DIG—digoxygenin; FU—fluorouracil; GPS—multi-gene proliferation signature; HR—hazard ratio; ISH—in situ hybridization; IRS—immunoreactive score; Lbs—laboratories; L(P)I—labeling (proliferation) index; LN—lymph node metastasis; MA—multivariate analysis; mAb—monoclonal antibody; meta—metastasis; MIB-1—antibody against Ki-67 antigen; MSI-H/L—microsatellite instability high/low; MSS—microsatellite stable; nd—not determined; no.—number; OS—overall survival; (p)CRC—(primary) colorectal cancer; PFS—progression-free survival; PL—Poland; pTNM—pathological tumor/node/metastasis; pTR—pathological tumor response; pts—patients; RC—rectal cancer; RFS—relapse-free survival; RR—relative risk; RT—radiotherapy; qRT-PCR—quantitative real-time polymerase chain reaction; RT-PCR—reverse transcriptase-polymerase chain reaction; S—surgery; semiq—semiquantitative; UA—univariate analysis; UICC—International Union Against Cancer; UK—United Kingdom; yr(s)—year(s).

**Table 4 cancers-15-04570-t004:** Prognostic value of selected long non-coding RNAs (lncRNAs) in colorectal cancer (CRC).

Type of lncRNA	Material/Research Model	Expression Level	Findings	Ref. No.
LOC285194	CRC (81); CRC cell lines: CaCO-2, HCT8, LoVo and C (CCC-HIE-2 cells); qRT-PCR	↓	A poor DFS; an independent predictor of DFS (MA)	[303]
PVT1	Pairs of CRC and C (164); CRC cell lines: RKO and HCT116; siRNA transfection; cell proliferation and invasion assays; gene expression array; qRT-PCR; array-CGH and copy no. analysis; gene set enrichment analysis; WB	↑	Promoted cell proliferation; a poor prognosis; an independent risk factor for OS (UA and MA)	[275]
	Pairs of CRC and C (210); qRT-PCR	↑	A shorter DFS and OS; an independent predictor of poor prognosis (MA)	[304]
DANCR	CRC (104); qRT-PCR	↑	A shorter OS and DFS; an independent poor prognostic factor for both OS and DFS (MA)	[305]
HOTTIP	CRC (156), C (21); qRT-PCR	↑	An unfavorable as well as an independent poor prognostic factor (MA)	[306]
SNHG20	CRC and C (107)	↑	An independent poor prognostic factor for OS (MA)	[316]
BANCR	CRC (106), C (65), qRT-PCR	↑	A shorter OS; an independent poor prognostic factor (HR 2.24, 95% CI 1.22–4.16)	[307]
SPRY4-IT1	CRC (106); qRT-PCR	↑	A poor OS; an independent prognostic factor (HR 2.34, 95% CI 1.14–4.82)	[308]
CCAT1 and CCAT2	CRC (280) and C (20); qRT-PCR	↑	A poor RFS and OS	[309]
XIST	CRC (196); CRC cell lines: LOVO, HT-29, HCT8, HCT116, SW480, and DLD1 and C (HCoEpics cells); qRT-PCR	↑	Could predict PFS and OS; could act as independent risk factor for poor prognosis	[310]
LINC00858	Pairs of CRC (115); CRC cell lines: T-29, HT-15, SW837 and SW1463; qRT-PCR; siRNA transfection; cell proliferation and apoptosis assays; colony formation assay; dual luciferase reporter assays; RIP; WB	↑	An independent poor prognostic factor	[284]
	Pairs of CRC (50) and 20 female BALB/c nude mouse; qRT-PCR; ISH; MTT assay; BrdU staining; FCM, wound healing, and Transwell assays; luciferase activity assay and RIP; IHC; WB; HE staining	↑	Prognostic factor for OS	[277]
MIR22HG	CRC (79) and C (84); CRC cell lines LoVo and HCT116; bioinformatics screen; qRT-PCR; MTT and Transwell assays; mouse model	↓	A poor OS and DFS; promoted cell survival, proliferation and tumor meta in vitro and in vivo	[287]

Legend: ↑,↓—high (upregulation), low (downregulation); BANCR-BRAF—activated nc RNA; BrdU—bromodeoxyuridine/5-bromo-2’-deoxyuridine; C—control; CCAT1/2—colon cancer-associated transcript 1/2; CGH array—comparative genomic hybridization array; CI—confidence interval; CRC–colorectal cancer; DANCR—anti-differentiation ncRNA; DFS—disease free survival; FCM—flow cytometry; HR—hazard ratio; HNF1A-AS1—HNF1A antisense RNA 1; HE—hematoxylin and eosin; HOTTIP—HOXA transcript at the distal tip; IHC—immunohistochemistry; ISH—in situ hybridization; MA—multivariate analysis; meta—metastasis; MIR22HG—MIR22 host gene; no.—number; OS—overall survival; PFS—poor progression-free survival; PVT1—plasmacytoma variant translocation 1; SNHG20—small nucleolar RNA host gene 20; SPRY4-IT1—sprouty RTK signaling antagonist 4-intronic transcript 1; RIP—RNA immunoprecipitation; qRT-PCR—quantitative real-time polymerase chain reaction; UA—univariate analysis; WB—Western blot analysis; XIST—X-inactive specific transcript.

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
