# Peer review of "Prognostic Biomarkers of Cell Proliferation in Colorectal Cancer (CRC): From Immunohistochemistry to Molecular Biology Techniques"

_cancers, 2023, doi:10.3390/cancers15184570_

Round 1

Reviewer 1 Report (New Reviewer)

Kasprzak A et al. has made an interesting study investigating the prognostic biomarkers of cell proliferation in colorectal cancer (CRC) by using the molecular techniques and immunohistochemistry. The present diligent study actually has set a good example for the followers who are endeavoring to looking for new biomarkers in the context of translational research. Furthermore, their experimental results were fruitful with a well written manuscript, and therefore I have no major criticisms. However, the present study was retrospective and thus liable to some uncertainty. Moreover, the concept of this article is not new, and the originality is questionable.

In order to compensate for the inherent flaws of the present study, I would like to suggest the citation of another related articles from the Asian fellow researchers with some discussion to more clarify this important issue:

1. Asian Journal of Surgery Volume 44, Issue 5, May 2021, Pages 715-722

Author Response

Dear Reviewer,

I wish to thank you very much for a review, and time spent on reviewing the manuscript.

Thank you very much for your valuable comments concerning my publication.

The current review concerned with the main methods of assessing cell proliferation and the most common markers of proliferation with prognostic significance in CRC (not all prognostic markers in CRC). I wanted to point out that the list of such markers is growing according to the development of research techniques, still used as complementary techniques, in conjunction with IHC, ISH, RT-PCR, etc.

As you suggested, I have cited the paper by Park et al, Asian J Surg, 2021, 44 (5): 715-722 (ref. no. 326) and added a small comment on this publication (subchapter 4.4.4. and 5) and ref. no. in the list of references.

Thank you for all the positive comments regarding the manuscript and I hope that the corrections will be positively accepted.

All suggested changes are marked in the text of the article.

Sincerely yours,

Aldona Kasprzak

Reviewer 2 Report (New Reviewer)

Summary / significance: The manuscript by Kasprzak is a comprehensive review of the literature on the field of biomarker detection in the context of colon cancer. I congratulate the author on the huge effort. The role of using biomarkers for analysing CRC not only on classical morphological but also on a non-invasive and therapeutic level is becoming increasingly important. This review is informative for both the expert and the beginner in the field. The author gives an introductory overview of cell proliferation mechanisms and their influence on the onset of colon cancer. The author particularly points out the diverse outcomes of the different studies regarding colon cancer risk prognosis. Therefore, an understanding of their detailed and subtle differences is essential and deserves further research.

Level of interest/merit: Understanding proliferative features of cancer is key for the identification of novel powerful markers and tools for use in CRC diagnostics. This is an interesting review focussing on studies that analyze the predictive power of CRC markers in clinical context.

Comment: The information provided in this article is extremely high and comprehensive, also summarized in instructive Tables and one Figure. The literature covered is huge, and the level of background is very high. The question arises whether the article does not exceed usual length. Several points could be addressed to make it more easy to read and some headers could be improved.

Specific comments:

1) Some headings and figure legends could be more explicit and detailed. For example, line 783 subchapter 4.3.4.1. and line 826, subchapter 4.3.4.2. have very short titles, maybe extend to “in CRC”. On the other hand, subchapter title 4.4, line 923, is extensively long, and instead maybe some introductory words to these methods could be added.

2) Some abbreviations are not explained when used for the first time, e.g. LI, Ts, Tpo  (lines 456ff), they could be written out when mentioned the first time.

The English level is quite high, however many phrasings need improvements: often the word order is inappropriate and some wordings are more colloquial. I have uploaded an edited pdf version with a number of proposed changes. If possible, the article should be edited and proofread by an English native speaker or expert.

Author Response

Dear Reviewer,

I wish to thank you very much for a review, and time spent on reviewing the manuscript.

As you suggested, the article has been completely verified by a native speaker and such a certificate is attached to Editor. I also took into account some of your suggestions attached in your PDF version, for which I am extremely grateful. I corrected the headings in some subsections and changed the subheading to subsection 4.4. to a shorter one. As suggested, subsection 4.4 currently contains text carried over from other parts of the paper regarding the methods that were used to study „non-immunocytochemical markers” of CRC. I have supplemented this section with an additional citation regarding methods to study gene methylation in CRC (ref. no. 278).

The abbreviations Li, Ts, and Tpot were already explained when they were first used. Nevertheless, thank you for your attention, it is now page 9 line 444-445.

As for the length of the paper, the description was first concerned with the main methods of assessing cell proliferation and the most common markers of proliferation with prognostic significance in CRC. The growing number of molecular techniques and the increase in the number of prognostic markers demonstrated by them obliged me to signal this group of factors as well. These were also the recommendations of other reviewers. Still, it is all about methods to assess cell proliferation and markers of prognostic significance in CRC.

Thank you for all the positive comments regarding the manuscript and I hope that the substantive and linguistic corrections will be positively accepted.

I've marked all changes to your comments.

Sincerely yours,

Aldona Kasprzak

Reviewer 3 Report (New Reviewer)

The author has done an outstanding job of thoroughly reviewing the topic of interest and the writing and organization are excellent.

I have a few comments regarding the MS:

1) The topic area is concerned with the older literature and history of proliferation biomarkers; as such, the review doesn't delve deeply into the more current biomarkers of CRC and which involve/reflect other processes such as apoptosis, senescence, TME, immune status of tumors, etc.

2) Similarly, it seems that liquid-based biopsies will be important to CRC classification and treatment responses, and these methods along with use of serum/plasma RNAs, DNA, proteins are important to bring into the picture (these areas are not sufficiently dealt with in the review).

3) The current version of the review, while excellent, is in my opinion more appropriate for a Pathology journal.    

Author Response

Dear Reviewer,

I wish to thank you very much for a review, and time spent on reviewing the manuscript.

Concerning comments on the manuscript:

  1. The aim of my current review was to show methods of assessing cell proliferation and prognostic markers of proliferation with prognostic significance in CRC. It was not the aim to show all markers of prognostic importance, but those most important in the everyday clinic of CRC. The involvement of markers in processes such as apoptosis, TME, CRC immune status are reviewed by other authors, e.g., the review by Koulis C, et al., Cancers, 2020 that I cited (ref. no. 323).

2.      I agree with your opinion regarding liquid-based biopsies that such material also provides data on the prognosis in CRC, but due to space limitations in the current review, I could not expand on it. However I signaled this problem in section 4.4.4, citing several recent papers (347-353), including the above review by Koulis C et al., 2020 (ref. no 323).    

3. Thank you for your kind attention, however, the publisher has accepted my previously submitted abstract for the current special issue. I hope that such an explanation will be positively received and that all the above explanations and any necessary changes will be accepted. 

I have highlighted all the changes in the text of the article.

Sincerely yours,

Aldona Kasprzak

Reviewer 4 Report (New Reviewer)

Well-compiled and well-written review article.

Author Response

Dear Reviewer,

We wish to thank you very much for a review, and time spent on reviewing the manuscript. Thank you once more and I really appreciate such a favourable review of my article.

The best regards, Aldona Kasprzak

Round 2

Reviewer 3 Report (New Reviewer)

MS is through and well organized.

This manuscript is a resubmission of an earlier submission. The following is a list of the peer review reports and author responses from that submission.

Round 1

Reviewer 1 Report

In this review manuscript, the author reviewed the results on the prognostic value of various cell cycle-related markers in colorectal cancer. There are some major concerns, as outlined below:

1. The manuscript has a heavy focus on the use of IHC to determine cell proliferation markers, but there are also many limitations associated with this technique that have led to contradictory reports. While it is important to provide a thorough overview of IHC, I believe it would be beneficial for the author to include more information on other novel molecular techniques that may be used as prognostic markers for CRC. In addition to the described RT-PCR and in situ hybridisation, the author should consider including findings from cell proliferation markers e.g. microRNA or Long non-coding RNA through RNA sequencing, DNA mutations through next generation sequencing, or DNA methylation through methylation profiling array. This could help to provide a more comprehensive and balanced understanding of the current state of research on this topic. I recommend that the authors consider simplifying the IHC section and focusing more on alternative molecular techniques and their potential advantages and drawbacks in the manuscript.

2. All the tables appear to contain a large number of cases without any grouping or summarization, which may make it difficult for readers to interpret the data effectively. I recommend considering ways to group or summarize the information in the table to make it more accessible and understandable for readers.

3. Any unpublished data or information should not be included in the review.

 4. It would be beneficial for the authors to consider using a figure to simplify the idea of various pathways for colon carcinogenesis. 

There are several grammatical mistakes throughout the manuscript which need to be addressed. Additionally, some of the sentences are quite lengthy and complex, which can make them difficult to follow. Simplifying the language and breaking up longer sentences may help improve clarity. 

Author Response

I wish to thank you very much for a review, and time spent on reviewing the manuscript. Thank you and I really appreciate such a favorable review of my work.

  1. Thank you very much for this comment and recommendation to include in this review paper, also the prognostic role of non-coding RNA in CRC. As recommended, I have added a whole subsection, although not at the expense of IHC markers, which I consider important, simple to perform, practical, and improving in the sense of at least evaluating IHC reactions. All techniques have their limits, and should be used complementarily. In addition, however, such a large number of papers ( including review papers) on ncRNAs is a bit depressing to once again choose something new and interesting. Therefore, I have chosen only exemplary ncRNAs that are related to the topic of the paper (proliferation, prognosis in CRC), also in Table 7.

  1. Thank you for this comment. Following the reviewer's advice, I have simplified the Tables slightly in terms of the data included. I added comments near the Tables regarding the criterion for their elaboration (a given marker in a given subsection, year of publication, technical details important for data comparison, the most important results obtained, etc.) and a short summary of the results obtained (which were mostly in the text). However, I did not want to duplicate detailed data in the text and in the Tables, but out of respect for the researchers, a summary of their research results was also included. I apologise if this was not very clear. In Tables 1-6, almost all the most important papers in the subject of a given IHC marker in the CRC over the years are included.

  1. I am sorry that I had to discard Figure 1 almost in its entirety. The unpublished data (microphotographs) came from my own research and here the service had only for illustration. Following the reviewer's recommendation, they were removed from Figure 1. I left only the expression of Ki-67 in CRC and cited which of our published papers they came from.

  1. It was not the purpose of my work to show signaling pathways in CRC, as this is a well-known topic and has been described in hundreds of review papers. In the present work, they are mentioned only in the context of activation of proliferation (Chapters 2 and 3) and as "targets" for non-coding RNA. The latter are in many figures in the papers I cited. I did not want to duplicate such figures, beautifully presented already. The purpose of the paper was primarily to present the contribution of immunocytochemistry and the simplest molecular bilogy techniques in the search for markers of cellular proliferation in CRC with prognostic significance. Please kindly accept this explanation.

The work is always corrected for language by the journal editors. Many times, despite our best efforts, still finds a lot of errors. However, according to the recommendation of the reviewer, the manuscript was corrected by a qualified, native speaker, familiar with the manuscript topics.

All changes (and all additions) in the text were marked red.

Reviewer 2 Report

Cancers:

 Prognostic biomarkers of cell proliferation in colorectal cancer (CRC): from immunohistochemistry to molecular techniques

 Aldona Kasprzak

The review is focused on the information available on current molecular tumor biomarkers and on establishing an optimal non-invasive prognostic and predictive test in CRC. There are biomarkers for the overall survival in resected colorectal liver metastases, such as cyclin B1, cyclin D1, proliferating cells, PCNA, and Ki-67. The current article reviews the most common cell cycle-related antigens assessed by different methods, including immunohistochemical, RT-PCR, or in situ hybridization as prognostic and predictive markers in CRC.

 PROS

The review is well-written and elaborate. The content is well presented in tabular form.

 CONS

Minor

Font size and spacing line 170-183.

Figure 1: The author should cite the source of the figure.

Can the authors explain if these citations have been involved within the elaborate references?

 Citation: Jørgensen, Lars N., et al. "Ki67 expression in stage II colorectal cancer: assessment of intra-tumoral heterogeneity and impact on prognosis." Virchows Archiv 471.6 (2017): 809-819.

 Citation: Zavrides, Harris, et al. "Cyclin D1 overexpression in colorectal carcinoma: correlation with clinicopathological features and prognosis." Anticancer Research 33.8 (2013): 3251-3256.

 Citation: Elsaleh, Hany, et al. "TP53 mutations in colorectal cancer from Iraq and Australia." World Journal of Gastroenterology 11.23 (2005): 3512-3517.

 Citation: Teodoridis, John M., et al. "Epigenetic inactivation of the p53-induced long noncoding RNA TP53 target 1 in human cancer." Proceedings of the National Academy of Sciences 112.27 (2015): E2828-E2836.

The over all flow is good.

Author Response

Thank you very much for the completed review, your time and overall positive comments. Thank you, once more. I have corrected the font size in the indicated lines and provided a citation at Fig. 1 (in blue color). It is currently truncated to only 2 photographs, just those from our already published research.

As for the indicated papers that the highly respected reviewer excerpted, unfortunately I did not cite any of them, as I simply neither earlier when writing the paper, nor now - I did not find them in PubMed. The title of the last paper is in the PubMed database, but the paper is about other authors and the year of publication is different, that is: Diaz-Lagares A, Crujeiras AB, Lopez-Serra P, et al. Epigenetic inactivation of the p53-induced long noncoding RNA TP53 target 1 in human cancer. Proc Natl Acad Sci U S A. 2016 Nov 22;113(47):E7535-E7544. doi: 10.1073/pnas.1608585113. Epub 2016 Nov 7. PMID: 27821766; PMCID: PMC5127373. This paper, however, does not directly link to my work, as it demonstrated the poor prognosis of one of the lncRNAs, but in gastric cancer, not CRC. Also this paper could not be cited. I apologize if I disappointed expectations.

Round 2

Reviewer 1 Report

I believe that the review could benefit from the inclusion of more recent findings and references. Therefore, the manuscript could be improved by incorporating more recent and advanced molecular techniques, such as DNA mutations through next-generation sequencing and DNA methylation through methylation profiling array.

It is not usual practice to include one's own published work in a review paper, as this can create potential conflicts of interest and bias in the review. Additionally, the authors should consider including the results of immunohistochemistry (IHC) for the proliferative markers discussed in the manuscript.

It would be beneficial for the reader to have a more simplified and integrated figure that summarizes the various pathways of colon carcinogenesis. This would greatly aid the reader in understanding the concepts without the need to refer to the cited review.

Minor editing of English language required.